# Combining systems and synthetic biology for in vivo enzymology

Sara Castaño-Cerezo [1✉], Alexandre Chamas [2,5,6], Hanna Kulyk [1,3], Christian Treitz [4], Floriant Bellvert[1,3], Andreas Tholey[4], Virginie Galéote[2], Carole Camarasa[2], Stéphanie Heux[1], Luis F Garcia-Alles[1], Pierre Millard [1,3✉] & Gilles Truan [1✉]

## Abstract

**Enzymatic parameters are classically determined in vitro, under conditions that are far from those encountered in cells, casting doubt on their physiological relevance. We developed a generic approach combining tools from synthetic and systems biology to measure enzymatic parameters in vivo. In the context of a synthetic carotenoid pathway in *Saccharomyces cerevisiae*, we focused on a phytoene synthase and three phytoene desaturases, which are difficult to study in vitro. We designed, built, and analyzed a collection of yeast strains mimicking substantial variations in substrate concentration by strategically manipulating the expression of geranyl-geranyl pyrophosphate (GGPP) synthase. We successfully determined in vivo Michaelis-Menten parameters ($K_M$, $V_{max}$, and $k_{cat}$) for GGPP-converting phytoene synthase from absolute metabolomics, fluxomics and proteomics data, highlighting differences between in vivo and in vitro parameters. Leveraging the versatility of the same set of strains, we then extracted enzymatic parameters for two of the three phytoene desaturases. Our approach demonstrates the feasibility of assessing enzymatic parameters directly in vivo, providing a novel perspective on the kinetic characteristics of enzymes in real cellular conditions.**

**Keywords** Carotenoid Synthesis; In Vivo vs In Vitro; Metabolism; Biotechnology; Enzymatic Parameters
**Subject Categories** Biotechnology & Synthetic Biology; Metabolism

## Introduction

Enzymes catalyze most of the chemical reactions in living systems. A comprehensive interpretation of enzymatic parameters is therefore of paramount importance to grasp the complexity of cellular metabolism. The vast amount of data collected for thousands of enzymes has contributed to significant progress in our understanding of the remarkable chemical capabilities of biocatalysts and of their roles in cellular reactions.

Enzymatic reactions are traditionally analyzed in vitro, under dilute conditions, using pure or semi-pure protein samples in buffer solution. In contrast, the cellular medium is most accurately viewed as a heterogeneous, dense, crowded gel, containing various types of macromolecules and cellular lipidic organelles, with potential partitioning effects and variations in substrate and/or product diffusion coefficients (McGuffee and Elcock, 2010). The parameters determined in classical enzymology experiments may therefore not be representative of in vivo reaction rates and equilibrium constants (Karen van Eunen, 2014; Ringe and Petsko, 2008). While some progress has been made in implementing and understanding viscosity and crowding effects in in vitro enzymatic assays, these conditions do not mimic the intrinsic complexity of the cellular environment (Karen van Eunen, 2014).

In vivo enzymology seems to be the obvious approach to measure enzymatic parameters inside cells. Early attempts were made using the enzyme photolyase, for which both in vitro and in vivo parameters were determined (Sancar, 2008; Harm et al, 1968). Meanwhile, the in vivo $V_{max}$ values of ten central carbon metabolism enzymes determined in the early 90 s revealed significant differences between in vivo and in vitro assays for heteromeric protein complexes (Wright et al, 1992). The past ten years have seen growing interest from systems biology in determining the in vivo $k_{cat}$ of native enzymes in model organisms. The in vivo apparent $k_{cat}$ of *E. coli* enzymes have been determined independently by two research groups, leveraging advances in absolute protein quantification and high throughput metabolomics (Heckmann et al, 2020; Davidi et al, 2016). These values were obtained by dividing the flux of enzymatic reactions by the absolute abundance of the corresponding enzymes. Both studies found correlations between in vivo and in vitro data (with correlation coefficients around 0.6), suggesting that this method could serve as an alternative to in vitro assays. While this systems biology approach has provided valuable information, it cannot be employed to determine apparent $K_M$s in vivo. Zotter et al measured the activity and affinity of TEM1-β lactamase in mammalian cells in vivo with confocal microscopy, using a fluorescently tagged

[1]TBI, Université de Toulouse, CNRS, INRAE, INSA, Toulouse, France. [2]SPO, Université Montpellier, INRAE, Institut Agro Montpellier, Montpellier, France. [3]MetaboHUB-MetaToul, National Infrastructure of Metabolomics and Fluxomics, Toulouse, France. [4]Systematic Proteome Research and Bioanalytics, Institute for Experimental Medicine, Christian-Albrechts-Universität zu Kiel, Kiel, Germany. [5]Present address: Department of Microbial Pathogenicity Mechanisms, Leibniz Institute for Natural Product Research and Infection Biology (HKI), Jena, Germany. [6]Present address: Cluster of Excellence Balance of the Microverse, Friedrich Schiller University Jena, Jena, Germany.
✉E-mail: castanoc@insa-toulouse.fr; millard@insa-toulouse.fr; truan@insa-toulouse.fr

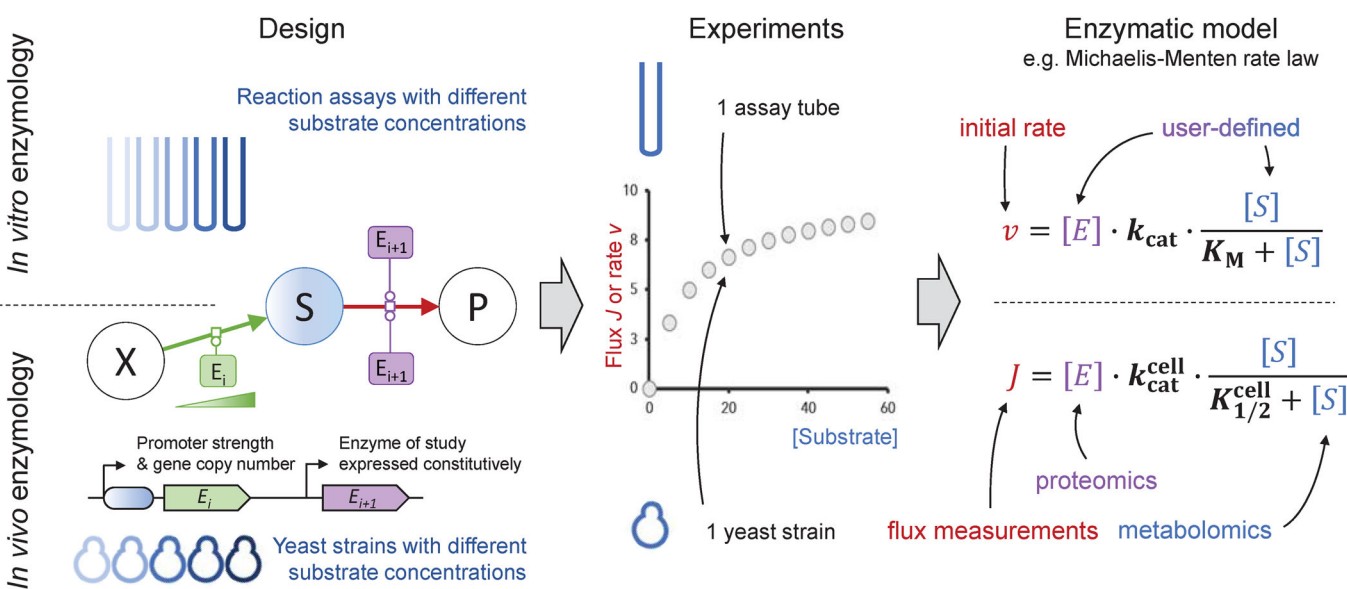

**Figure 1.** General principle of the proposed strategy for in vivo enzymology and comparison between in vitro and in vivo enzymology approaches.

enzyme and a fluorescent substrate and product (Zotter et al, 2017). They observed that the catalytic efficiency ($k_{cat}/K_M$) of this enzyme differs in vitro and in vivo due to substrate attenuation, indicating that in vitro data are not always indicative of in vivo function. Zotter et al is probably the most comprehensive in vivo enzymology study to date, but this approach cannot be generalized because of a lack of universally appropriate fluorescent substrates/products for all enzymes. In a recent investigation of the in vivo kinetic parameters of thymidylate kinase (TmK), an interesting finding was the difference in TmK's activity pattern when the substrate (thymidine monophosphate, TMP) was supplied in the media versus provided by internal metabolism, with Michaelis–Menten kinetics in the former and Hill-like kinetics in the latter. The authors hypothesized that the limited diffusion of TMP might be due to its confinement in a putative metabolon in *E. coli* (Bhattacharyya et al, 2021).

In vivo enzymology is a particularly attractive prospect for membrane and multimeric proteins (Wright et al, 1992), which are tedious to purify and for which activity assays are difficult to optimize, mostly because artificial membranes are required (Takakuwa et al, 1994; Camagna et al, 2019; Urban et al, 1994; Gemmecker et al, 2015; Iwata-Reuyl et al, 2003). In the industrially important carotenoid pathway for example, despite the expression of numerous carotene biosynthesis enzymes, our understanding of their enzymatic behavior remains limited because many are membrane-associated proteins. While kinetic parameters for some phytoene synthases, sourced from plants or bacteria, have been established in vitro, many enzymatic assays require the co-expression of geranylgeranyl pyrophosphate (GGPP) synthase to attain activity, possibly because of the amphiphilic nature of the substrate or the requirement of membranes (Camagna et al, 2019; Iwata-Reuyl et al, 2003). Another example of the difficulty of in vitro assays is phytoene desaturase, for which discernible enzyme activity has only been achieved in engineered environments (e.g., liposomes) with cell-purified substrate (Gemmecker et al, 2015;

Schaub et al, 2012; Fournié and Truan, 2020; Stickforth and Sandmann, 2011).

This study combines synthetic and systems biology tools to develop an original in vivo enzymology approach. Synthetic biology offers a remarkable set of genetic engineering tools to precisely modulate the activity of enzymes in synthetic pathways, enabling in vivo control of substrate concentrations for the studied enzymatic reactions. In turn, systems biology can provide quantitative data on these reactions (fluxes and enzyme, substrate and product concentrations) and computational tools to model their behavior. We applied this approach to investigate a synthetic carotenoid production pathway in *Saccharomyces cerevisiae*, with industrial applications ranging from foods to pharmaceuticals.

## Results

### General principle of the proposed in vivo enzymology method

Enzymes are usually characterized in terms of their affinity ($K_M$) and activity (maximal reaction rate $V_{max}$ and turnover number $k_{cat}$) (Fig. 1). These parameters are typically determined in vitro by varying the substrate concentration across a relatively broad range (about two orders of magnitude) and measuring the reaction rate for each substrate concentration (Michaelis and Menten, 1913). Different mathematical formulas, such as the Michaelis–Menten equation, have been derived to estimate enzymatic parameters from these data. We propose a novel approach wherein the substrate concentration is varied directly within cells (Fig. 1), by modulating the concentration of the enzyme producing the substrate of the studied reaction. Levels of the substrate-producing enzyme are varied using different combinations of promoter strengths and gene copy numbers, and translate into a wide range of substrate concentrations (2–3 orders of magnitude, as detailed in the

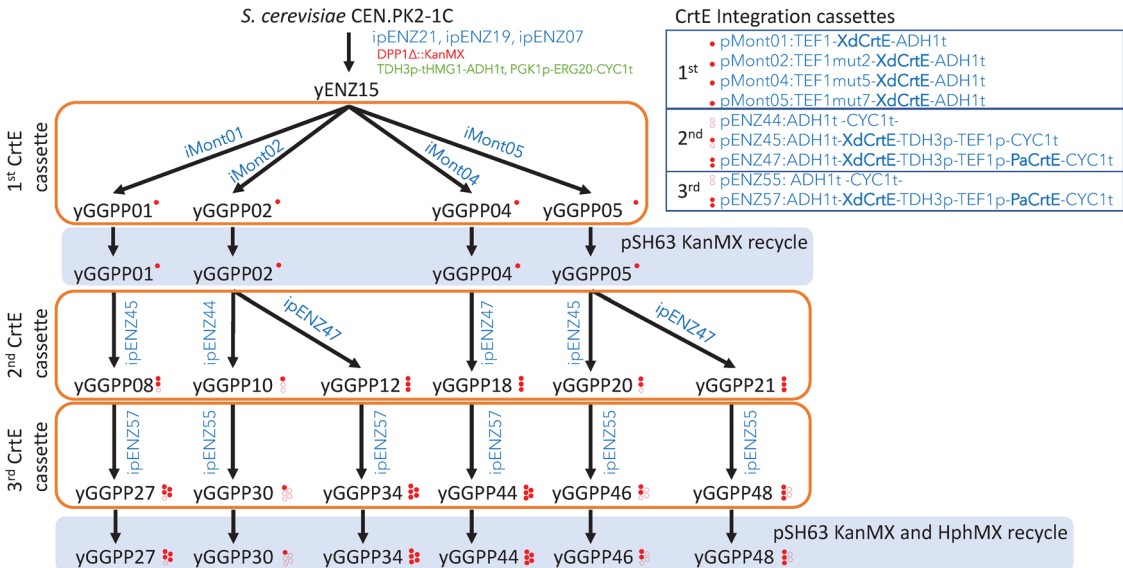

**Figure 2. Scheme of the construction of yeast strains with different GGPP concentrations.**

Red filled dots represent a copy of GGPP synthase, and empty dots represent an empty integration cassette.

following sections). In our experimental setup, each substrate concentration is achieved using a specifically engineered yeast strain. In each of these strains, the gene encoding the enzyme of interest is expressed at a given level, mirroring the conditions of in vitro experiments. Engineered strains are then grown under steady-state conditions (i.e., exponential growth), where reaction fluxes and metabolite concentrations remain constant (Bruggeman et al, 2007). This stability allows for the accurate measurement of product formation fluxes (equivalent to reaction rates), intracellular substrate concentrations, and enzyme concentrations. The data obtained can then be combined to calculate apparent in vivo kinetics parameters, denoted $K_{1/2}^{cell}$, $V_{max}^{cell}$ and $k_{cat}^{cell}$ in analogy with the classical $K_M$, $V_{max}$, and $k_{cat}$ parameters (Fig. 1).

## Construction of yeast strains to investigate a synthetic carotenoid production pathway

We evaluated our strategy by investigating a synthetic carotenoid pathway in yeast (*S. cerevisiae*), an industrially important chassis in biotechnology, which includes two membrane-interacting enzymes that are challenging to investigate in vitro (Camagna et al, 2019; Schaub et al, 2012; Neudert et al, 1998; Fournié and Truan, 2020): phytoene synthase (CrtB) and phytoene desaturase (CrtI). Phytoene synthase, the first enzyme in the carotenoid pathway and considered the bottleneck of carotenoid biosynthesis in plants (Zhou et al, 2022), condenses two molecules of geranylgeranyl pyrophosphate (GGPP) head-to-head to form phytoene. Phytoene synthases have proved difficult to express in soluble and active forms in microorganisms (Camagna et al, 2019; Neudert et al, 1998; Iwata-Reuyl et al, 2003). In vitro rates and affinity constants have been determined for both plant and bacterial phytoene synthases but these parameters may have been altered by the conditions of the in vitro assays—adding detergents for the bacterial enzyme and co-expressing a GGPP synthase or using semi-crude extracts for the plant enzyme (Camagna et al, 2019; Iwata-

Reuyl et al, 2003; Neudert et al, 1998; Fraser et al, 2000; Camara, 1993; Schofield and Paliyath, 2005). Following this enzymatic step, three insaturations are introduced in phytoene to produce lycopene. In non-photosynthetic bacteria and fungi, three reaction steps are catalyzed by a single enzyme (CrtI in bacteria and CarB in fungi) (Sandmann, 2009), whereas in plants and cyanobacteria, four different enzymes are involved in the conversion of phytoene to lycopene (Koschmieder et al, 2017). Given that these desaturation steps are a major bottleneck in microbial carotenoid biosynthesis, determining in vivo enzymatic constants is an industrially relevant challenge (Iwata-Reuyl et al, 2003; Rabeharindranto et al, 2019; Chen et al, 2016).

To analyze these enzymes, we assembled a set of strains covering a broad range of substrate concentrations (Fig. 2 and Table EV1). To create the collection of strains with different intracellular levels of GGPP, the substrate of phytoene synthase, we first increased GGPP production by modifying the native terpene pathway: strong constitutional expression of *ERG20* and of the truncated form of *HMG1* and deletion of the main phosphatase gene involved in GGPP and FPP pyrophosphate group removal (*DPP1*), yielding the strain yENZ15 (Westfall et al, 2012; Zhao et al, 2017). Three different integration cassettes containing different microbial GGPP synthases (CrtE) expressed under different constitutive promoters were integrated into the yENZ15 genome, yielding a series of strains designed to produce a wide range of GGPP concentrations (Fig. 2).

We then verified that the intracellular GGPP concentration was indeed modulated in vivo, as done previously (Schofield and Paliyath, 2005; Peng et al, 2018; Chambon et al, 1990; Tokuhiro et al, 2009). GGPP concentration varied by a factor of 38 in these strains (Fig. 3B), and the strains all had similar growth rates $(0.40 \pm 0.03\ h^{-1})$ (Fig. 3B) indicating that there was no change in cell physiology. As detailed below, we used this set of strains as a base to study a phytoene synthase (https://www.uniprot.org/uniprotkb/P21683/entry, EC 2.5.1.32) and three different lycopene-forming phytoene desaturases (EC 1.3.99.31).

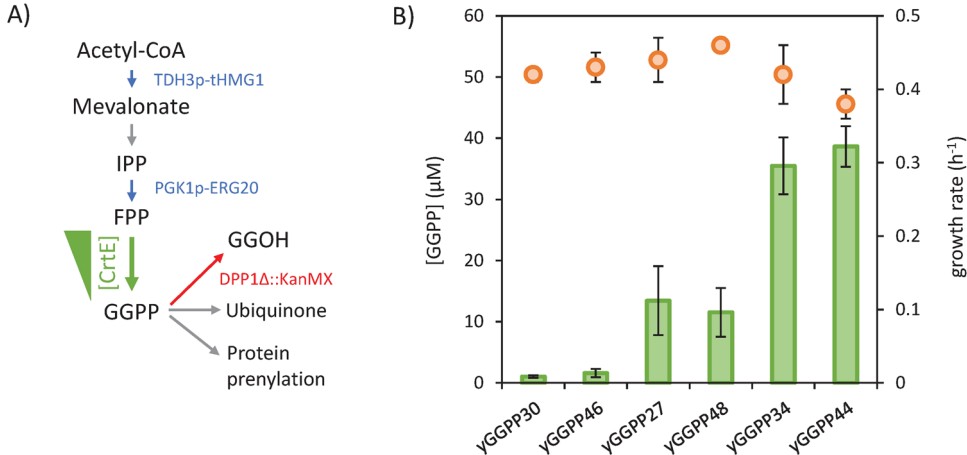

**Figure 3.    Construction of the set of yeast strains with different intracellular GGPP concentrations.**

(**A**) Genomic modification scheme for the yENZ15 strain and intracellular fates of GGPP in *S. cerevisiae*. Overexpressed enzymes are shown in blue, enzyme deletion is shown in red, and enzyme modulation is shown in green. (**B**) GGPP concentration (green bars) and specific growth rate of *S. cerevisiae* strains (orange dots). Mean values ± standard deviations (error bars) were estimated from three independent biological replicates. Source data are available online for this figure.

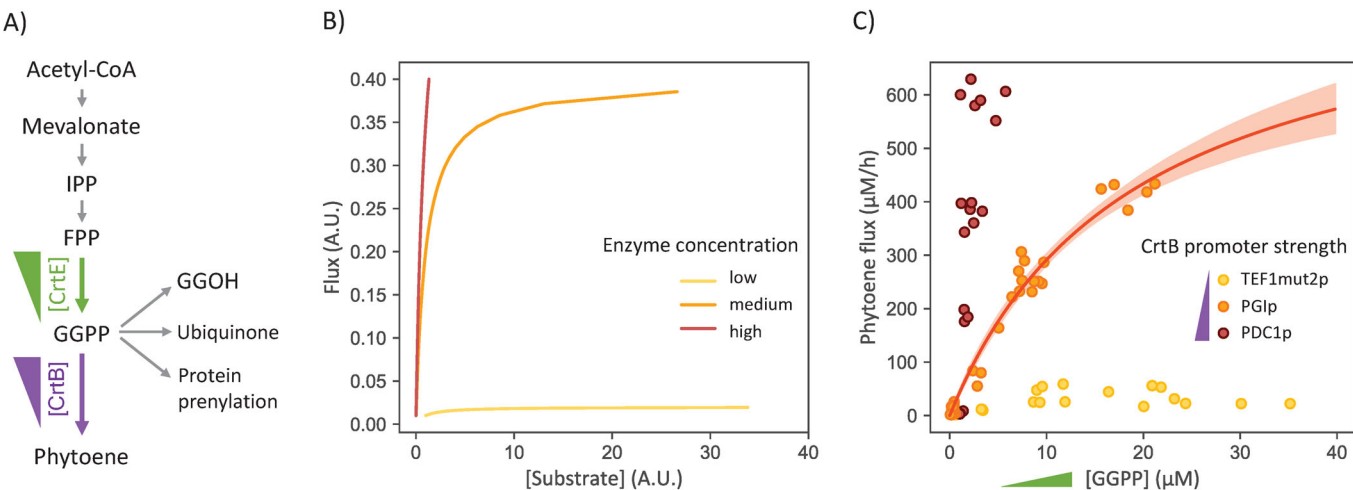

**Figure 4.    In vivo characterization of phytoene synthase from *Pantoea ananas*.**

(**A**) Scheme for the biosynthesis of phytoene by CrtB. (**B**) Steady-state flux and substrate concentration simulated for three different CrtB activities (low, medium and high) under a broad range of GGPP-producing flux. (**C**) In vivo characterization of CrtB in *S. cerevisiae*: phytoene production as a function of GGPP concentration in strains with low (TEF1mut2p), medium (PGIp) and high (PDC1p) level of CrtB. Each data point represents an independent biological replicate. For PGIp strains, red line represents the best fit of a Michaelis-Menten rate law, and shaded area corresponds to 95% confidence interval on the fit. Source data are available online for this figure.

## In vivo enzymatic parameters of phytoene synthase from *Pantoea ananas*

We used the set of strains with different intracellular levels of GGPP (Fig. 3) to study phytoene synthase from *Pantoea ananas* (PaCrtB) (Fig. 4A). In a classical enzymatic reaction setup, enzyme concentration must be carefully controlled to establish an informative set of reaction rate vs substrate concentration data points. To evaluate the impact of enzyme concentration on the collected kinetic profiles, we developed a kinetic model and performed simulations for three different enzyme concentrations (low, medium and high) under a broad range of substrate-

producing flux. The relationships between the steady-state substrate concentration and the flux are shown in Fig. 4B. Simulation results suggest that excessively high enzyme concentrations would hinder substrate saturation, rendering parameter estimation impossible. In contrast, an extremely low enzyme concentration would lead to a low production flux, making precise measurements difficult and reducing the chance of obtaining accurate enzyme parameters. Therefore, we decided to perform our saturation experiments using three enzyme concentrations by expressing *P. ananas crtB* under the control of three different constitutive promoters (low expression: TEF1mut2p, medium expression: PGI1p and high expression: PDC1p).

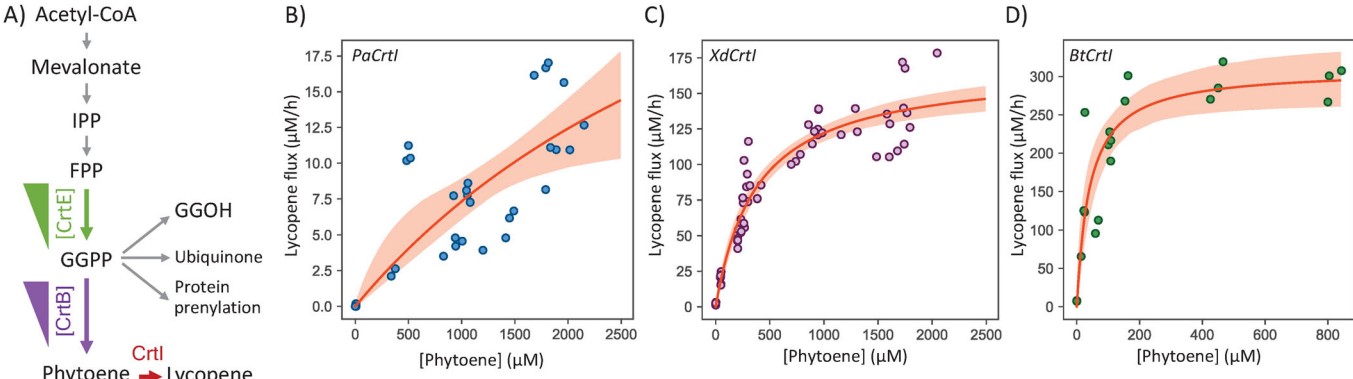

**Figure 5.  In vivo characterization of three phytoene desaturases.**

(A) Scheme for the biosynthesis of lycopene by CrtI. (B–D) In vivo characterization of CrtI from *P. ananas* (B), *X. dendrorus* (C), and *B. trispora* (D). Each data point represents an independent biological replicate, red lines represent the best fits of a Michaelis-Menten rate law, and shaded areas correspond to 95% confidence intervals on the fits. Source data are available online for this figure.

Consistent with simulation results (Fig. 4B), the GGPP concentration progressively decreased as PaCrtB activity increased (Fig. 4C). Therefore, the maximal GGPP concentration was very low at high PaCrtB level (PDC1p), and enzyme saturation was not reached. In contrast, the phytoene flux increased as PaCrtB expression increased. The phytoene concentration was too low for accurate quantification of the phytoene flux at low levels of PaCrtB (TEF1mut2p). However, accurate measurements of GGPP concentration and phytoene flux could be made at medium PaCrtB levels (PGIp). Based on these results, we used strains with medium PaCrtB levels for further investigations. A 167-fold variation in GGPP concentration (from 0.12 to 20.78 μM) was observed in these strains. Partial saturation was achieved, leading to a nonlinear relationship between the phytoene flux and the GGPP concentration, as observed in classical in vitro experiments. We assumed and verified, by western blot and mass spectrometry-based proteomics, that PaCrtB levels were identical in all strains (Fig. EV1 and Table EV2), indicating that PaCrtB activity ($V_{max}^{cell}$) was also similar in all strains. Precise estimates of $V_{max}^{cell}$ (849 ± 71 μM/h, rsd = 13%) and $K_{1/2}^{cell}$ (19 ± 3 μM, rsd = 14%) were obtained by fitting these data using an irreversible Michaelis-Menten model (consistent with previous reports (Neudert et al, 1998) and with the very negative $\Delta G^0$ value of this reaction) (Table EV3). As expected from the simulations, parameters could not be determined precisely from the two other datasets (Table EV3). The affinity of PaCrtB for GGPP estimated from our in vivo data is higher than measured in vitro ($K_M$ = 41 μM) (Neudert et al, 1998), highlighting the necessity of measuring enzyme parameters directly within the intracellular environment. Meanwhile, the $k_{cat}^{cell}$ of 6 ± 1 s$^{-1}$, obtained from absolute quantitative proteomics measurements of the enzyme concentration (42 nM) in strains expressing PaCrtB under the control of PGI1p, is comparable to the one obtained in vitro for CrtB in *Pantoea agglomerans* (14 s$^{-1}$, Table EV4) (Iwata-Reuyl et al, 2003).

The theoretical maximum phytoene production flux in these strains is therefore 849 ± 71 μM/h. The bottleneck for carotenoid biosynthesis in plants is phytoene synthase, whose activity is regulated by a combination of transcriptional, post-transcriptional, and post-translational mechanisms to adjust carotenoid production

(Zhou et al, 2022). In our synthetic system however, our results suggest that GGPP biosynthesis is the limiting step since five copies of the GGPP synthase gene are required to achieve just 51% of the maximum phytoene production flux, due to the lack of phytoene synthase saturation.

## Straightforward extension to analyze downstream enzymes of the carotenoid pathway: phytoene desaturases

The most time-consuming tasks in the proposed in vivo enzymology protocol are the molecular cloning and strain engineering steps required to generate strains with different substrate concentrations. However, the strains used to investigate a given enzyme can be recycled to investigate downstream steps in the same metabolic pathway, significantly streamlining experiments, which become a simple plug-and-measure process (Fig. 5A). Integration of *crtB*, controlled by the high-expression promoter PDC1 into our initial set of strains, produced strains with a 430-fold variation in phytoene concentration (from 4 to 1641 μM). These strains thus form an ideal framework to investigate the in vivo kinetics of the next metabolic step with various phytoene desaturases.

To demonstrate this point, we chose three widely used microbial lycopene-forming phytoene desaturases (1.3.99.31): *P. ananas* CrtI (P21685, PaCrtI), which, along with *Rubrivivax gelatinosus* CrtI, is one of only two lycopene-forming phytoene desaturases with published in vitro enzymatic parameters (Schaub et al, 2012; Stickforth and Sandmann, 2011); and two fungal CrtIs widely used in carotenoid production, one from *Phaffia rhodozyma*, formely *Xanthophyllomyces dendrorhous* (A0A0P0KMF9, XdCrtI), and the other from *Blakeslea trispora* (Q67GI0, BtCrtI), with no enzymatic data available to date.

As for the investigation of phytoene synthase, the expression of the phytoene desaturases was controlled by different promoters: TDH3p for PaCrtI, PGK1p for XdCrtI, and PGI1p for BtCrtI. These promoters were chosen based on the relative lycopene producing abilities of the different CrtIs in *S. cerevisiae*. This was especially important for BtCrtI, where high expression in the exponential phase caused cell death due to the formation of lycopene crystals

(Fig. EV2A). PaCrtI and XdCrtI levels did not vary with the phytoene concentration of the strains, but BtCrtI levels were slightly higher in the strains with the lowest phytoene concentrations (Fig. EV3). The higher levels of PaCrtI compared with those of the fungal phytoene desaturases is explained in large part by the use of a TDH3 promoter. However, lycopene fluxes were 10 times lower with PaCrtI than with XdCrtI or BtCrtI (Fig. 5). Phytoene levels were high enough to saturate both fungal CrtIs (Fig. 5) but not PaCrtI, for which lycopene fluxes remained low for all tested phytoene concentrations.

The experimental data from the three enzymes were fitted with an irreversible Michaelis-Menten equation, from which $V_{max}^{cell}$ and $K_{1/2}^{cell}$ values were successfully obtained for XdCrtI and BtCrtI but not for PaCrtI, for which only lower limits for $V_{max}^{cell}$ (13 μM·h⁻¹) and $K_{1/2}^{cell}$ (560 μM) could be determined (Table EV3). The $V_{max}^{cell}$ values obtained for BtCrtI (309 ± 22 μM·h⁻¹) were higher than for XdCrtI (168 ± 7 μM·h⁻¹), and the $K_{1/2}^{cell}$ for BtCrtI (41 ± 12 μM) was 9 times smaller than XdCrtI's (380 ± 49 μM). The $k_{cat}^{cell}$ obtained by integrating quantitative proteomics measurements in strains expressing XdCrtI under PGK1p was 0.030 ± 0.006 s⁻¹ (Table EV3).

These results highlight the versatility of our approach to compare the in vivo behavior of enzymes from different organisms, using the same set of strains, and provide valuable information to understand and optimize natural and synthetic pathways in vivo.

# Discussion

Inspired by conventional in vitro enzyme assays, we developed an innovative approach to measure enzyme kinetics in vivo. We successfully used the proposed method to estimate the in vivo equivalents of Michaelis-Menten parameters for a phytoene synthase and two phytoene desaturases in S. cerevisiae.

These results highlight the reliability and versatility of our in vivo enzymology approach, which hinges on solving two methodological challenges: (1) controlling the substrate pool over a broad concentration range, and (2) measuring the total concentrations of substrate, product, and enzyme. The first challenge was addressed using synthetic biology tools to modulate the concentration of a substrate-producing enzyme (here GGPP synthase, CrtE). The second challenge was overcome thanks to sensitive, quantitative metabolomics and proteomics techniques that can be generalized to various other enzymatic models. Indeed, the coverage of the metabolome and fluxome by current omics approaches is now high and is continuously increasing, which enables the application of the proposed approach to a broad range of enzymes. In this study, we quantified three basic enzymatic parameters ($K_{1/2}^{cell}$, $k_{cat}^{cell}$ and $V_{max}^{cell}$). In the absence of absolute quantitative proteomics data, the proposed method still allows for the determination of $V_{max}^{cell}$ and $K_{1/2}^{cell}$ values, providing sufficient information for most enzymology and metabolic engineering studies. Similarly, estimating the absolute $K_{1/2}^{cell}$ values can be achieved using only relative flux values, without requiring absolute concentration of enzymes.

We demonstrate that substrate concentrations can be varied in vivo over a wide range. Here, the GGPP concentration was varied by a factor 167 and the phytoene concentration by a factor 430, allowing clear enzyme saturation curves to be observed for the two fungal phytoene desaturases (XdCrtI and BtCrtI). However,

while the range of substrate concentrations explored was significantly wider than those used in in vitro studies of the same enzymes (Table EV4) (Schaub et al, 2012; Neudert et al, 1998), complete saturation was not achieved for PaCrtB or PaCrtI. It is conceivable that, when these enzymes are bound to natural membranes, their apparent affinity for the substrate could differ from the in vitro environment. A second explanation concerns substrate availability. While classical in vitro enzyme assays involve homogeneous, highly diluted buffers, the intracellular environment is dense, heterogeneous and compartmentalized. In this first in vivo approach, variations in substrate concentrations between compartments were not considered explicitly, though it may have affected the values obtained for the parameters. For example, a higher concentration of phytoene inside lipid droplets compared with the rest of the membrane would reduce phytoene concentrations around CrtI enzymes and would thus limit substrate saturation. Enzyme localization could also affect measurements as conditions (pH, concentration of ions, etc.) vary between compartments. Iwata-Reuyl et al found for instance that the measured activity of PaCrtB is 2000 times lower in the absence of detergents (Iwata-Reuyl et al, 2003), and Fournié et Truan observed that different heterologous CrtI expression systems produced different phytoene saturation patterns (Leys et al, 2003). In the case of the two P. ananas enzymes, the mechanisms that cause the absence of saturation may be different. For PaCrtI, the observed concentrations of phytoene (substrate) and lycopene (product) are either equivalent to or lower than those obtained with the two fungal CrtI enzymes. This suggests a true difference in the affinity for phytoene between the bacterial and fungal enzymes. For PaCrtB, given that the GGPP concentration in the cell is high (around 20 μM), the lack of complete saturation is more likely due to a heterogeneous distribution of GGPP within the cell, with only a fraction of GGPP being in the vicinity of the enzyme. Dedicated studies will be required to clarify the effects of cell compartments on enzyme efficiency.

In addition, other explanations for the difficulty in reaching complete saturation merit attention as they also convey general hypotheses on metabolic systems: (i) substrate or product toxicity detrimental to cell growth (e.g., the specific growth rate is nearly 50% lower for lycopene concentrations above 400 μM, (Fig. EV2B), (ii) overflow mechanisms that relocate some of the substrate to a different cell compartment or even outside the cell (Phégnon et al, 2024; Park et al, 2016), (iii) the presence of alternative pathways that divert additional substrate produced above a certain concentration threshold (Bu et al, 2022; Zhu et al, 2017), or (iv) intrinsic properties of metabolic systems whereby production fluxes tend to decrease when product concentrations increase (Heinrich and Rapoport, 1974; Kacser and Burns, 1973; Enjalbert et al, 2017). While these mechanisms contribute to global metabolite homeostasis (Fendt et al, 2010; Millard et al, 2017; Reaves et al, 2013) and are essential for cell viability, they limit substrate accumulation and thus could prevent complete saturation of the enzyme. Moreover, it is crucial to bear in mind that, similarly to in vitro data, enzymatic parameters are only valid under the conditions used to measure them. We therefore argue that enzymatic parameters should not be considered constants since they depend on the microenvironment (pH, ion concentration, temperature, membrane composition, etc.), be that in vitro or in vivo. Still, as mentioned below, these parameters are useful for pathway engineering and an advantage of our in vivo enzymology method is that kinetics parameters are measured under the exact same conditions as the enzyme is expressed.

Our approach enables the determination of whether a given enzyme is operating at saturation in vivo under different cellular conditions. The tested enzymes displayed various saturation profiles, indicating that they may function under diverse conditions within the cell. Operating at full saturation (far above the $K_{1/2}^{cell}$, the enzyme works at its maximum rate, being unaffected by changes in substrate concentration or minor environmental variations. This makes the enzymatic reaction highly robust in terms of product formation. Conversely, if an enzyme operates at substrate concentrations much lower than the $K_{1/2}^{cell}$, any variation in substrate concentration will directly affect the production rate. This could maintain substrate homeostasis, potentially prevent toxic effects due to concentration changes. From a metabolic engineering perspective, achieving the optimal balance between high product formation and metabolic homeostasis is crucial. This balance can be attained by adjusting the levels of both the substrate-forming enzyme and the target enzyme. This process needs to be repeated for each enzyme, potentially under different saturation regimes. In biotechnology, our in vivo enzymology approach may thus guide metabolic engineering strategies to ensure that the overall pathway maintains the desired balance between production and cellular homeostasis, thereby ensuring greater stability and robustness of the engineered microbial strains at maximal production flux.

The final step in our in vivo enzymology method involves fitting experimental data with a mathematical model of enzyme kinetics to obtain the corresponding parameters, the same approach as used in vitro. Traditional enzymatic models were derived with in vitro measurements and assumptions in mind. For instance, the classical Michaelis-Menten relationship applies only at steady-state, a condition clearly met in our experimental setup where all data were collected during exponential growth (i.e., metabolite concentrations and fluxes remain constant over time). However, other assumptions do not hold in vivo, in particular regarding the product concentration, which cannot be zero. Nevertheless, for the enzymes investigated here, where the catalyzed reactions are essentially irreversible, the reverse reaction can be neglected, and the Michaelis-Menten formalism still applies. Another important assumption is that the substrate concentration must be higher than the enzyme concentration. In this study, the in vivo substrate/enzyme ratio was between 8 and 1400 for PaCrtB and between 2 and 2500 for XdCrtI (Table EV5). Surprisingly, this criterion was not met for CrtB and CrtI in previous in vitro assays (Michaelis and Menten, 1913; Phégnon et al, 2024), with substrate/enzyme ratios between 0.05 and 0.18 for PaCrtB and between 1 and 2.57 for PaCrtI (Table EV4). In our setup, despite some formal assumptions not being fully satisfied, the data were still satisfactorily fit with a Michaelis-Menten equation. This underscores the applicability of our method, from which informative parameters such as the degree of saturation can be inferred to understand in vivo enzyme function and to engineer natural and synthetic pathways for biotechnology. The availability of in vivo data about enzyme kinetics may also lead to the derivation of specific laws that account for the specificities of in vivo studies (e.g., non-negligible product concentrations).

As we advocate for the simplicity of our method, we would like to share some insights on its implementation in future studies. First, to maximize the output of genetic constructs, experimental design should be employed to compare various enzymes that catalyze the same reaction (whether mutants or from different species) or to target multiple enzymes in the same metabolic pathway, in this case phytoene synthase and phytoene desaturase. Second, the number of genetic constructs necessary to reach saturation (ideally 4 to 5) can be minimized by verifying the substrate production range early on. Third, in most cases, testing three enzyme concentrations has been sufficient to obtain a satisfactory saturation curve or at least to precisely estimate the $K_{1/2}^{cell}$ range. Lastly, improvements can be made using high-throughput methods for the construction of plasmids and strains, for growth experiments and for samples preparation steps. The development of single cell metabolomics and proteomics approaches will also significantly increase the throughput of the strain characterization step in future studies.

Sixteen years ago, in their review on enzyme function, Dagmar Ringe and Gregory Petsko wrote: "How do enzymes function in a crowded medium of low water activity, where there may be no such thing as a freely diffusing, isolated protein molecule? In vivo enzymology is the logical next step along the road that Phillips, Koshland, and their predecessors and successors have traveled so brilliantly so far" (Ringe and Petsko, 2008). The work presented here, albeit performed in the context of a synthetic metabolic pathway, touches on the difference between kinetic parameters measured in vitro and in vivo and their interpretation. We notably show how fine-tuning and balancing the expression of the substrate-producing enzyme and the enzyme under study yields datasets from which meaningful and reliable enzymatic parameters ($K_{1/2}^{cell}$, $k_{cat}^{cell}$ and $V_{max}^{cell}$) can be obtained. By including additional controlled steps, this method could be applied to a wider range of variables, such as inhibition and activation parameters (by modulating the pool of regulatory metabolites). This method could also benefit from dynamic data (time-course monitoring in response to metabolic or genetic perturbations). New formalisms will be required to account for in vivo conditions, notably the presence of products, similar enzyme and substrate concentrations, local concentration variations, and molecular fluxes within the cell. Our method is particularly valuable for studying membrane-bound and multimeric enzymes, for which the purification and assay optimization steps of classical in vitro enzymology can be extremely challenging. For membrane-bound enzymes, in vivo enzymology offers a realistic environment devoid of detergents or other interferences, with natural membranes rather than liposomes. We sincerely hope that our work will stimulate further studies delving deeper into how enzymes function in their natural environment.

## Methods

**Reagents and tools table**

| Reagent/Resource | Reference or Source | Identifier or Catalog Number |
|---|---|---|
| **Experimental Models** | | |
| *Saccharomyces cerevisiae* CEN.PK2-1C | EUROSCARF | 30000A |
| **Recombinant DNA** | | |
| pMRI34-CrtE-Gal1-10-HMG1t | (Xie et al, 2014) | N/A |
| YEplac195 YB/I | EUROSCARF | P30796 |

| Reagent/Resource | Reference or Source | Identifier or Catalog Number |
|---|---|---|
| pAC-BETA | Addgene | 53272 |
| pCfB2903 | Addgene | 73275 |
| TEF1p | (Nevoigt et al, 2006) | N/A |
| TEF1p_mutant plasmid 2 | (Nevoigt et al, 2006) | N/A |
| TEF1p_mutant plasmid 4 | (Nevoigt et al, 2006) | N/A |
| TEF1p_mutant plasmid 5 | (Nevoigt et al, 2006) | N/A |
| TEF1p_mutant plasmid 6 | (Nevoigt et al, 2006) | N/A |
| TEF1p_mutant plasmid 7 | (Nevoigt et al, 2006) | N/A |
| TEF1p_mutant plasmid 10 | (Nevoigt et al, 2006) | N/A |
| **sAntibodies** | | |
| FLAG Tag Monoclonal Antibody | ThermoFisher Scientific | MA1-91878 |
| V5 Tag Monoclonal Antibody | ThermoFisher Scientific | MA5-15253 |
| Goat anti-Mouse IgG (H + L) Secondary Antibody, HRP | Invitrogen | A16066 |
| **Oligonucleotides and other sequence-based reagents** | | |
| Primers | IDT | Table EV7 |
| Synthetic DNA | Twist Bioscience | Dataset EV1 |
| AQUA peptides | ThermoFisher Scientific | Table EV8 |
| **Chemicals, Enzymes and other reagents** | | |
| Hifi Gibson assembly | NEB | E2621 |
| Phusion High-Fidelity DNA Polymerase | ThermoFisher Scientific | F530L |
| Phire Green Hot Start II PCR Master Mix | ThermoFisher Scientific | F126 |
| DNA release | ThermoFisher Scientific | F355 |
| GGPP | Cayman chemical | 63330 |
| Phytoene | CaroteNature | No. 0044 |
| Lycopene | Sigma-Aldrich | SMB00706 |
| trans-β-Apo-8´-carotenal | Sigma-Aldrich | 10810 |
| G418 | Formedium | G4185 |
| Hygromycin | Formedium | HYG1000 |
| Complete CSM | Formedium | DCS0019 |
| Lithium acetate | Sigma-Aldrich | L4158 |
| SuperSignal West Pico PLUS substrate | ThermoFisher Scientific | 34577 |
| Bradford assay | ThermoFisher Scientific | 23246 |
| HeavyPeptide AQUA Ultimate | ThermoFisher Scientific | Custom order |
| **Software** | | |
| Benchling | https://www.benchling.com/ | |
| Proteome Discoverer | ThermoFisher Scientific | v3.01.27 |

| Reagent/Resource | Reference or Source | Identifier or Catalog Number |
|---|---|---|
| COPASI | https://copasi.org/ | v4.39 |
| Python | https://www.python.org/ | v3.12 |
| **Other** | | |
| FastPrep FP120 cell disruptor | Thermo (Electron Corporation) | BIO 101 |
| Thermo Scientific Vanquish Focused UHPLC Plus system | Thermo Scientific | UHPLC |
| Diode Array Detector HL LightPipe 60 mm HighSens flow cell | Thermo Scientific | DAD |
| YMC carotenoid column (100 × 2.0 mm and 3 μm particle size) | YMC | CT99S03-1002WT |
| YMC precolumn (100 × 2.0 mm and 3 μm particle size) | YMC | CT99S03-01Q1GC |
| SC110A SpeedVac Plus | ThermoFisher | |
| Thermo Scientific Q Exactive Plus hybrid quadrupole-Orbitrap mass spectrometer | ThermoFisher Scientific | HRMS |
| Thermo Scientific Hypersil GOLD column 100 × 2.1 mm, 3 μm. | Thermo Scientific | 25003-102130 |
| Hypersil Gold 3 μm 10 × 2.1 mm DROP-IN GUARDS 4/PK | Thermo Scientific | 25003-012101 |
| Trans-Blot SD Cell | BioRad | 1703940 |
| UltiMate 3000 UHLC system | ThermoFisher Scientific | NCS-3500RS nano |
| PepMap100, 5 μm, 300 Å Trap Cartridge | ThermoFisher Scientific | 174500 |
| PepMap RSLC C18 column | ThermoFisher Scientific | 164570 |

## Plasmid construction

Plasmids and primers are listed in Tables EV6–7. Plasmid sequences and annotations are provided in Dataset EV1. The primers were synthesized by IDT (Leuven, Belgium) and the sequences of PaCrtB, PaCrtI, and BtCrtI were codon optimized for yeast and synthetized by Twist Bioscience (San Francisco, California). XdCrtE and XdCrtI from *Enterobacter agglomerans* were amplified from pMRI34-CrtE-Gal1-10-HMG1t, YEplac195 YB/I, and pAC-BETA respectively (Verwaal et al, 2007; Cunningham et al, 1996; Xie et al, 2014). Sequences of the mutated versions of the TEF1 promoter TEF1mut2p, TEF1mut5p, and TEF1mut7p were obtained from Nevoigt et al (2006). pCfB2903(XI-2 Marker-Free) was a gift from Irina Borodina (Jessop-Fabre et al, 2016). Polymerase chain reaction (PCR) was performed using Phusion high-fidelity polymerase and Phire Hot start II DNA polymerase (ThermoFisher Scientific, Lithuania). DNA fragments were purified using Monarch DNA Gel Extraction Kit from New England Biolabs. DNA fragments were annealed by isothermal assembly using NEBuilder HiFi assembly kit from New Englands Biolabs. Clones and plasmids were propagated in homemade calcium- and TOP10-competent *Escherichia coli* cells.

## Construction of yeast strains

All yeast strains used in this study are derived from CEN.PK2-1C and are listed in Table EV1. Yeasts were transformed using Gietz et al's high-efficiency transformation protocol (Gietz, 2014). Integrative cassettes were obtained by enzyme digestion or PCR and were used without any further purification. Strains were selected using auxotrophy markers or antibiotic resistance at a concentration of 400 μg/mL. Antibiotic resistance recycling was performed using vector pSH63 as described in the literature (Güldener et al, 1996). Genome integration was verified by colony PCR using the primers listed in Table EV7. Genomic DNA was extracted using DNA release from ThermoFisher Scientific.

## Media and culture conditions

All strains were grown in modified synthetic Verduyn media containing glucose (111 mM), $NH_4Cl$ (75 mM), $KH_2PO_4$ (22 mM), $MgSO_4$ (0.4 mM) and CSM (ForMedium LTD, Hunstanton, England) at pH 5.0 (Verduyn et al). Sterilization was performed by filtration. Fresh colonies from selective plates were precultured in 350 μL complete synthetic medium at 28 °C for 8 h and these cells were used to inoculate cultures with a 1:5 medium:flask proportion to an initial $OD_{600nm}$ of 0.002, grown at 200 rpm at 28 °C.

## Carotene quantification

Samples (10 or 20 mL) of yeast culture were harvested with an $OD_{600nm}$ of ~5, centrifuged, and washed with 1 mL of MilliQ water. Cell pellets were freeze-dried and stored at −80 °C until extracted. β-apocarotenal solution (40 μL, 50 μM) was added to the dried cells, and carotenes were extracted with glass beads and 500 μL of acetone in three 20 s rounds of agitation at 0.05 m/s with a FastPrep FP120 cell disruptor (ThermoFisher). The acetone phase was transferred to a new tube and the extraction was repeated twice. Acetone extracts were pooled, centrifuged, dried under nitrogen flux, and resuspended in acetone for HPLC analysis. Analyses were carried out on a Thermo Scientific Vanquish Focused UHPLC Plus system with DAD HL. Extract samples (5 μL) were injected into a YMC carotenoid column (100 × 2.0 mm and 3 μm particle size) equipped with a precolumn (100 × 2.0 mm and 3 μm particle size). The mobile phases used to separate and quantify phytoene, lycopene and β-apocarotenal from ergosterol and derivatives were mixtures of (A) methanol/water (95:5) and (B) dichloromethane. The flow was 0.25 mL/min with the following gradient: 0–0.1 min 5% B, 0.1–0.5 min 20% B, 0.5–2 min 60% B, 2–5 min 80% B, 5–8 min 80% B and 8–11 min 5% B. The absorbance from 210 to 600 nm was measured throughout the run with a data collection rate of 2 Hz and a response time of 2 s. The phytoene concentration was deduced from its absorbance at 286 nm and lycopene and β-apocarotenal concentrations from the absorbance at 478 nm. The reference wavelength (600 nm) was subtracted from each of the wavelengths used for metabolite quantification.

## Flux calculation

Phytoene and lycopene are produced and accumulate in the cells, and their pools are continuously diluted by cell growth. Assuming an absence of degradation or reutilization of these end-products by the cell, phytoene and lycopene production fluxes are balanced solely by their dilution fluxes in the exponential growth phase, where cells are at metabolic steady-state. Thus, phytoene and lycopene production fluxes were determined by multiplying their concentrations by the cell growth rate. This flux calculation method provides results consistent with those obtained by targeted $^{13}C$-fluxomics (Millard et al, 2020).

## GGPP quantification

GGPP was quantified as detailed previously (Castaño-Cerezo et al, 2019). Briefly, 10 mL of broth was filtered through 0.45 μm Sartolon polyamide (Sartorius, Goettingen, Germany) and washed with 5 mL of fresh culture medium (without glucose). The filters were rapidly plunged into liquid nitrogen and then stored at −80 °C until extraction. Intracellular GGPP was extracted by incubating filters in closed glass tubes containing 5 mL of an isopropanol/$H_2O$ $NH_4HCO_3$ 100 mM (50/50) mixture at 70 °C for 10 min. For absolute GGPP quantification, 50 μL of $^{13}C$ internal standard were added to each extract. Cellular extracts were cooled on ice and sonicated for 1 min. Cell debris was removed by centrifugation (5000 × g, 4 °C, 5 min). Supernatants were evaporated overnight (SC110A SpeedVac Plus, ThermoFisher, Waltham, MA, USA), resuspended in 200 μL of methanol:$NH_4OH$ 10 mM (7:3) at pH 9.5 and stored at −80 °C until analysis.

Analyses were carried out on a LC–MS platform composed of a Thermo Scientific Vanquish Focused UHPLC Plus system with DAD HL, coupled to a Thermo Scientific Q Exactive Plus hybrid quadrupole-Orbitrap mass spectrometer (Thermo-Fisher), as detailed previously (Castaño-Cerezo et al, 2019). Calibration mixtures (prepared at concentrations from 0.08 nM to 10 μM) were used to construct calibration curves from which the absolute concentration of GGPP in the samples was determined.

## Western blot

Protein extracts were prepared as described by Zhang et al (2011). Briefly, 1.5 $OD_{600nm}$ of pelleted cells were pre-treated with 100 μL of a 2 M lithium acetate cold solution, and left to stand for 5 min, followed by 5 min centrifugation at 5000 × g, 4 °C. The supernatant was discarded and 100 μL of a 0.4 M solution of NaOH was added. After gentle resuspension, and 5 min standing on ice, the samples were centrifuged for 5 min at 4 °C. After discarding supernatants, the pellets were vigorously vortexed with 60 μL of bromophenol blue loading dye solution with 5% β-mercaptoethanol. After denaturation for 10 min at 99 °C, 5 μL of each sample was loaded onto 10% SDS page gel. Semi-dry transfer was performed on PVDF membrane (Merck Millipore, Darmstadt, Germany) using a Trans-Blot SD Cell BioRad apparatus (18 V during 20 min), and 5% bovine milk in TBS as blocking agent. Incubations were performed with mouse anti-FLAG or mouse anti-V5 (Thermo-Fisher Scientific), and secondary anti-mouse IgG coupled with horseradish peroxidase (ThermoFisher Scientific), diluted according to manufacturer instructions. Proteins were detected by incubation with SuperSignal West Pico PLUS substrate (Thermo-Fisher Scientific).

## Proteomics

For cell disruption, $10^8$ cells were dissolved in 200 µL of lysis buffer (0.1 M NaOH, 2% SDS, 2% 2-mercaptoethanol, 0.05 M EDTA), heated at 90 °C for 10 min and neutralized with 5 µL of 4 M acetic acid. Glass beads were added and the samples were vortexed at 4 °C for 30 min. Cell debris was pelleted by centrifugation and $3000 \times g$ for 10 min and the protein concentration of the supernatant was determined using the Bradford assay. Protein aliquots (400 µg) were cleaned by methanol chloroform precipitation. The protein samples were dissolved in 5 µL of 6 M guanidinium chloride, 5 µL of 0.1 M dithiothreitol (DTT), and 100 µL of 50 mM TEAB 50 and diluted to a protein concentration of 2 µg/µL with 90 µL of water. Absolute protein quantification was performed using heavy isotope labeled tryptic peptides as internal standards. Protein lysate aliquots (20 µg) were spiked with a mixture of ten AQUA peptides, C-terminally labeled with heavy lysine or arginine (ThermoFisher Scientific) with concentrations of 50 fmol/µg, 10 fmol/µg, 5 fmol/µg, and 0 fmol/µg. The samples were reduced by adding 2 µL of 0.1 M DTT and incubating at 60 °C for 1 h, alkylated with 2 µL of chloroacetamide 0.5 M at room temperature for 30 min, and digested with 0.5 µg of trypsin overnight. Digestion was stopped by adding 5 µL of 1% TFA and the samples were cleaned using 100 µL C18 tips. The extracts were lyophilized, reconstituted in 20 µL of eluent A, and transferred to HPLC vials.

Samples were analyzed in triplicate using an UltiMate 3000 UHLC system coupled to a Q Exactive Plus mass spectrometer (ThermoFisher Scientific). Approximately 0.6 µg of peptides were loaded on a C18 precolumn (PepMap100, 5 µm, 300 Å, Thermo-Fisher Scientific) and separated on a PepMap RSLC C18 column (50 cm × 75 µm, 2 µm, 100 Å, ThermoFisher Scientific) with a 1.5 h gradient with eluent A (Water, 0.05% FA) and eluent B (80% ACN, 0.04% FA) with a flow rate of 0.3 µL/min. The peptides were first desalted for 4 min at 5% B, then separated with a gradient to 50% B over 65 min, to 90% B in 3 min, held at 90% B for 8 min, and then equilibrated for 8 min at 4% B.

For targeted absolute quantification, full MS spectra measurements were followed by parallel reaction monitoring (PRM) of targeted heavy AQUA peptides and the corresponding light, native peptide of the proteins of interest. The full MS spectra were acquired in profile mode with the following settings: resolution of 70,000, mass range of 350–950 $m/z$, AGC target of $10^6$, and 80 ms maximum injection time. PRM MS2 spectra were acquired with an isolation window of 1.6 $m/z$, 17,500 resolution, AGC target of $10^5$, 80 ms injection time, and an NCE of 27. The inclusion list contained 26 entries covering the light and heavy species of 10 peptides at the most intense charge state (2+ or 3+).

Database searches were performed with Proteome Discoverer (version 3.01.27; ThermoFisher). The raw data were compared with the UniProt protein databases of *S. cerevisiae* strain CEN.PK113-7D (UniProt 02.2023; 5439 entries), and common contaminants using Sequest HT and Chimerys search algorithms. The Sequest search parameters were a semi-tryptic protease specificity with a maximum of 2 missed cleavage sites. The precursor mass tolerance was 8 ppm and the fragment mass tolerance was 0.02 Da. Oxidation of methionine and acetylation of protein N-termini were allowed as dynamic modifications. Carbamidomethylation of cysteine was set as a static modification. Chimerys database searches were performed with default settings. Percolator q-values were used to restrict the false discovery

rate (FDR) of peptide spectrum matches to 0.01. The FDR of peptide and protein identifications was restricted to 1% and strict parsimony principles were applied to protein grouping.

A spectral library of the AQUA peptides was generated using Proteome Discoverer with a sample containing only 500 fmol of the ten AQUA peptides, in addition to the PRM analyses of the spiked yeast samples. The PRM data of three concentrations analyzed in triplicate were imported and all transitions were reviewed using Skyline (MacLean et al, 2010). Between four and nine transitions without interference were chosen for each peptide and the ratio of light and heavy peptides of the sum of transitions was calculated for absolute quantification. The list of used peptides and all corresponding transitions are provided in Table EV8.

## Calculation of absolute concentrations

Protein and metabolite concentrations are expressed as absolute intracellular concentrations. Metabolite concentrations initially expressed in µmol/g DCW were converted into absolute intracellular concentrations using a conversion factor of $6.59 \times 10^{10}$ cells/g DCW (Fig. EV4) and a cellular volume of 66 µm³ (Punekar, 2018). The intracellular concentrations of PaCrtB and XdCrtI were calculated using a conversion factor of 0.63 g protein/g DCW for yeast grown on ammonium sulfate medium (Albers et al, 1996). All calculations for data conversion are provided in the corresponding Extended view Tables.

## Modeling

To test the impact of enzyme expression level on the obtained kinetic profiles, we built a toy kinetic model of the pathway under study. This model contains two metabolites (GGPP and phytoene) and three reactions: GGPP formation by CrtE, defined as a constant flux; GGPP conversion into phytoene by CrtB, modeled using a Michaelis-Menten rate law; and phytoene dilution by growth, modeled using mass action. The $K_M$ value of CrtB was set arbitrarily to 1, and we simulated the steady-state phytoene production flux and GGPP concentration for three CrtB levels ($V_{max}$ set to 0.02, 0.4, and 0.7) under a broad range of GGPP-producing flux (from 0.01 to 0.4 µM/h). The model has been developed with COPASI v4.39 (Hoops et al, 2006) and is available from our GitHub repository (https://github.com/MetaSys-LISBP/in_vivo_enzymatic_parameters) and from the Biomodels database (Malik-Sheriff et al, 2020) under accession ID MODEL2407240001. All simulations were performed with COPASI.

## Calculation of enzyme parameters

Enzymatic parameters ($V_{max}$ and $K_M$) were determined by fitting a Michaelis-Menten equation to the measured relationships between substrate concentrations and reaction rates. Uncertainties on fitted parameters were determined using a Monte-Carlo approach. Briefly, 1000 simulated noisy datasets were generated (where the noise was determined as the standard deviation of residuals obtained for the fit of the experimental dataset), and the mean value, standard deviation, and 95% confidence intervals of each parameter were determined from the distribution of values obtained for the 1000 datasets. For each enzyme, we verified that all parameters were identifiable based on the covariance matrix and on the results of the Monte-Carlo analysis. The code for parameter estimation and statistical analysis is provided

as a Jupyter notebook at https://github.com/MetaSys-LISBP/in_vivo_enzymatic_parameters.

## Data availability

The data used to make the figures can be found in the Source data files. The model and computer code produced in this study are available in the following databases: Model: BioModels MODEL2407240001 (https://www.ebi.ac.uk/biomodels/MODEL2407240001). Code: GitHub (https://github.com/MetaSys-LISBP/in_vivo_enzymatic_parameters).

The source data of this paper are collected in the following database record: biostudies:S-SCDT-10_1038-S44318-024-00251-w.

## Peer review information

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

## Acknowledgements

The authors thank MetaboHub-MetaToul (Metabolomics & Fluxomics facilities, Toulouse, France, https://mth-metatoul.com), which is part of the French National Infrastructure for Metabolomics and Fluxomics (https://www.metabohub.fr), funded by the ANR (MetaboHUB-ANR-11-INBS-0010), for access to MS facilities. The authors also thank Jean-Charles Portais (RESTORE, Geroscience & Rejuvenation Center, Université de Toulouse, INSERM, CNRS, EFS, Toulouse, France), Sylvie Dequin (SPO, Université Montpellier, INRAE, Institut Agro Montpellier, Montpellier, France), Thibault Nidelet (SPO, Université Montpellier, INRAE, Institut Agro Montpellier, Montpellier, France), Thomas Lautier (TBI, Université de Toulouse, CNRS, INRAE, INSA, Toulouse, France) and Sergueï Sokol (TBI, Université de Toulouse, CNRS, INRAE, INSA, Toulouse, France) for insightful discussions. This work was supported by the French National Research Agency project ENZINVIVO (ANR-16-CE11-0022).

## Author contributions

**Sara Castaño-Cerezo**: Conceptualization; Data curation; Formal analysis; Supervision; Validation; Investigation; Visualization; Methodology; Writing—original draft; Writing—review and editing. **Alexandre Chamas**: Formal analysis; Investigation; Methodology; Writing—review and editing. **Hanna**

**Kulyk**: Formal analysis; Investigation; Methodology; Writing—review and editing. **Christian Treitz**: Formal analysis; Investigation; Methodology; Writing—review and editing. **Floriant Bellvert**: Methodology; Writing—review and editing. **Andreas Tholey**: Investigation; Methodology; Writing—review and editing. **Virginie Galéote**: Supervision; Investigation; Methodology; Writing—review and editing. **Carole Camarasa**: Conceptualization; Investigation; Methodology; Project administration; Writing—review and editing. **Stéphanie Heux**: Conceptualization; Funding acquisition; Methodology; Project administration; Writing—review and editing. **Luis F Garcia-Alles**: Formal analysis; Methodology; Writing—review and editing. **Pierre Millard**: Conceptualization; Software; Formal analysis; Supervision; Funding acquisition; Investigation; Visualization; Methodology; Writing—original draft; Writing—review and editing. **Gilles Truan**: Conceptualization; Formal analysis; Supervision; Funding acquisition; Visualization; Methodology; Writing—original draft; Project administration; Writing—review and editing.

Source data underlying figure panels in this paper may have individual authorship assigned. Where available, figure panel/source data authorship is listed in the following database record: biostudies:S-SCDT-10_1038-S44318-024-00251-w.

## Disclosure and competing interests statement

The authors declare no competing interests.

# Expanded View Figures

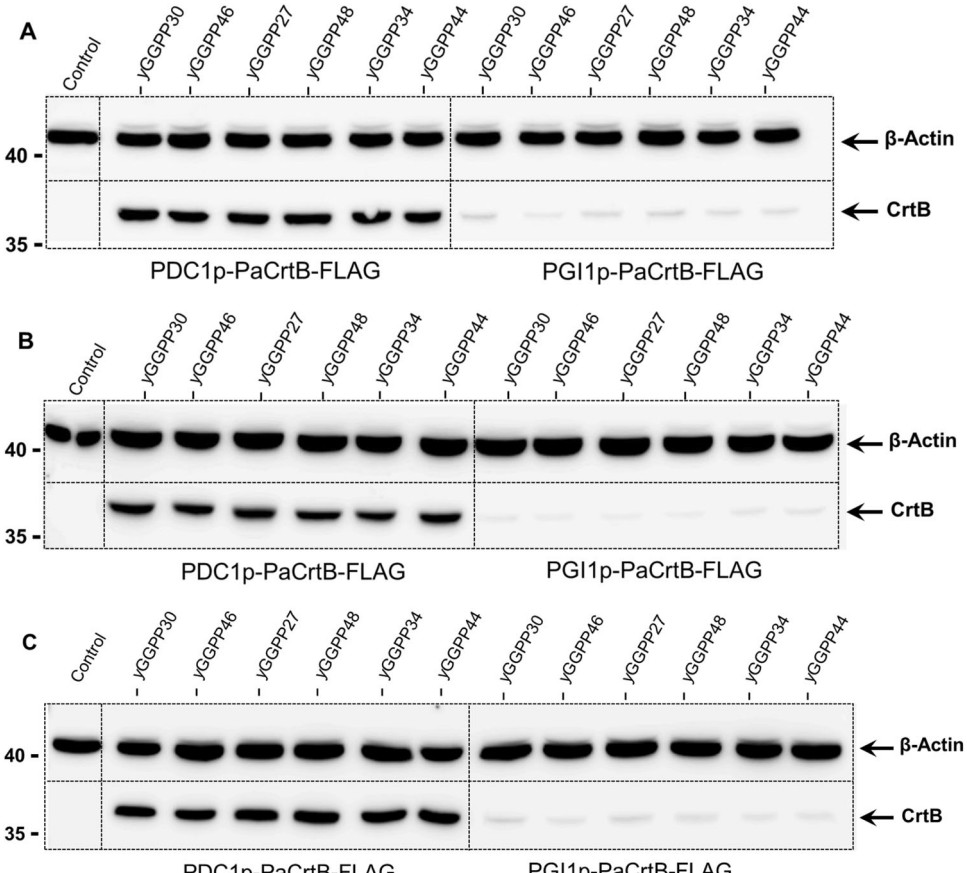

**Figure EV1.   Expression of PaCrtB-FLAG in the different strains.**

PaCrtB-FLAG expression determined by western blot in yeast strains with different GGPP concentrations (panels (**A**–**C**) correspond to three different biological replicates).

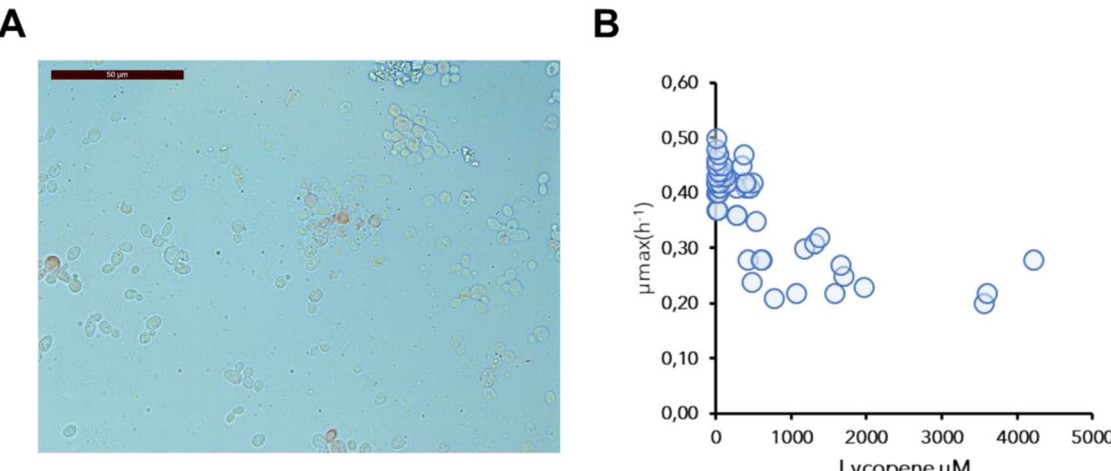

**Figure EV2. Lycopene production in *S. cerevisiae*.**

(A) *S. cerevisiae* yGGPP34 strain expressing PDC1p-PaCrtB and TDH3p-BtCrtI. Bright field images were acquired using the camera LEICA DFC300FX mounted in the microscope Leica DM4000B with Leica EL6000 light source. Lycopene crystals are observed in red. (B) Decrease of specific growth rate in yeast strains expressing BtCrtI with different phytoene concentrations.

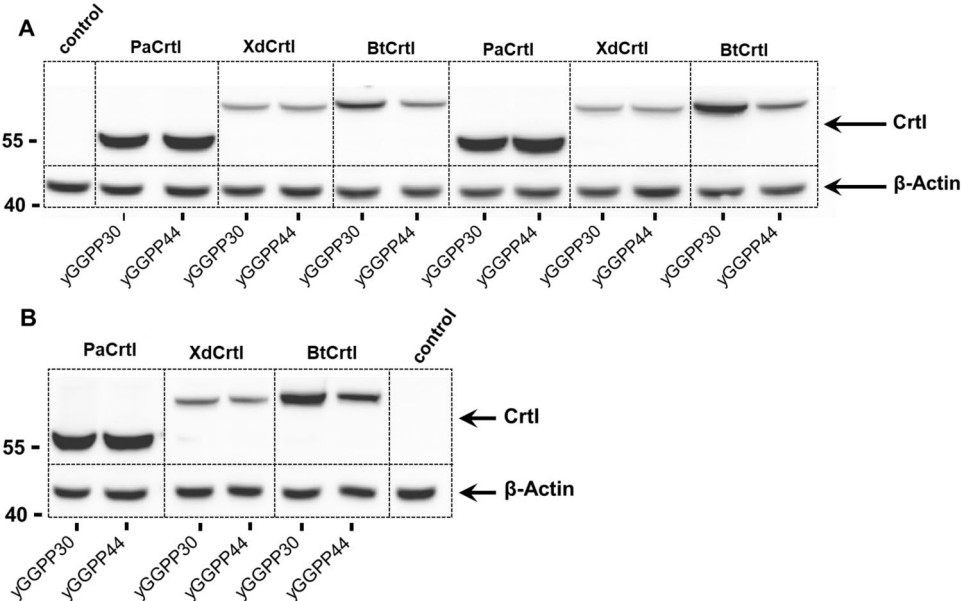

**Figure EV3. Expression of CrtI in the different strains.**

Expression of the three studied CrtI-V5 proteins in strains with low (yGGPP030) and high (yGGPP044) phytoene content (panels (**A** and **B**) correspond to three different biological replicates).

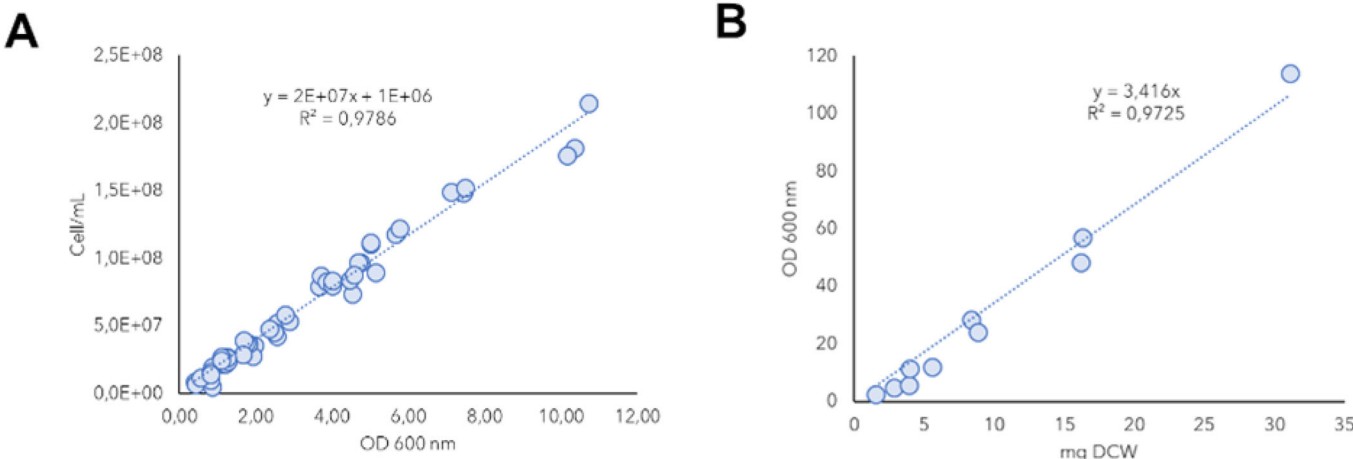

**Figure EV4. Correlations between OD and biomass concentration.**

Correlation between cell/mL and $OD_{600nm}$ (**A**) and between $OD_{600nm}$ and mg DCW (**B**).

