## [Peer Review File · The EMBO Journal]

Combining systems and synthetic biology for in vivo enzymology

Sara Castaño-Cerezo, Alexandre Chamas, Hanna Kulyk, Christian Treitz, Florian Bellvert, Andreas Tholey, Virginie Galeote, Carole Camarasa, Stéphanie Heux, Luis Garcia-Alles, Pierre Millard, and Gilles Truan

Corresponding author(s): Gilles Truan (gilles.truan@insa-toulouse.fr) , Pierre Millard (millard@insa-toulouse.fr), Sara Castaño-Cerezo (castanoc@insa-toulouse.fr)

Review Timeline:

Submission Date:	29th Feb 24
Editorial Decision:	8th Apr 24
Revision Received:	1st Aug 24
Editorial Decision:	3rd Sep 24
Revision Received:	11th Sep 24
Accepted:	12th Sep 24

Editor: Hartmut Vodermaier

Transaction Report:

Dr. Gilles Truan
Toulouse Biotechnology Institute
135 Avenue de Rangueil
Toulouse 31077
France

8th Apr 2024

Re: EMBOJ-2024-117139
Combining systems and synthetic biology for in vivo enzymology

Dear Dr. Truan,

Thank you for submitting your study on new systems and synthetic biology approaches for in vivo determination of enzyme parameters to The EMBO Journal. I sent the manuscript to three expert referees, who have now returned the reports copied below. Since all of them express interest in the work, we would be happy to consider a revised version further for publication, pending adequate addressing of various conceptual, presentational, and technical issues raised by the reviewers. Since it is our policy to consider only a single round of major revision and therefore important to fully respond to all comments at the time of resubmission, please do not hesitate to get back to me with a tentative response letter/revision plan, in case you would like to clarify/discuss certain points and how they might be answered already during the early stages of the revision. I should add that we could also offer extension of the default three-months revision period if needed, with our 'scooping protection' (meaning that competing work appearing elsewhere in the meantime will not affect our considerations of your study) remaining of course valid also throughout this extension.

Detailed information on preparing, formatting and uploading a revised manuscript can be found below and in our Guide to Authors. Thank you again for the opportunity to consider this work for The EMBO Journal, and I look forward to your revision in due time.

Yours sincerely,

Hartmut Vodermaier

9) Digital image enhancement is acceptable practice, as long as it accurately represents the original data and conforms to community standards. If a figure has been subjected to significant electronic manipulation, this must be clearly noted in the figure legend and/or the 'Materials and Methods' section. The editors reserve the right to request original versions of figures and the original images that were used to assemble the figure. Finally, we generally encourage uploading of numerical as well as gel/blot image source data; for details see: embopress.org/page/journal/14602075/authorguide#sourcedata

At EMBO Press, we ask authors to provide source data for the main manuscript figures. Our source data coordinator will contact you to discuss which figure panels we would need source data for and will also provide you with helpful tips on how to upload and organize the files.

In the interest of ensuring the conceptual advance provided by the work, we recommend submitting a revision within 3 months (7th Jul 2024). Please discuss the revision progress ahead of this time with the editor if you require more time to complete the revisions. Use the link below to submit your revision:

Link Not Available

Referee #1:

The paper describes what the authors call a generic approach for measuring enzymatic parameters (in this case, basic kinetic parameters) *in vivo*. It has been recognized for a long time that enzymes studies in dilute aqueous solution, the *de facto* condition for a hundred years, are in an environment unlike that inside a typical cell, where the molecular density approaches that of Times Square on New Year's Eve. (Ironically, the density inside a protein crystal - an environment scorned by classical enzymologists in the early days of structural biology as being completely artificial - is, in terms of crowding at least, more realistic.) As structural biology, driven by cryo-Electron Tomography, moves into the realm of *in vivo* observation, there is need for measurements of fundamental enzymatic parameters to do the same. That makes the work presented here potentially of great interest.

The authors use a number of steps to obtain their measurements, each of which has sufficient specific issues to render the claim of "a generic method" a bit exaggerated; nevertheless, the method is both clever and adaptable. The first step is the choice of organism: budding yeast is the best-studied eukaryote and one readily amenable to genetic manipulation and biochemical observation. The second, and cleverest, step is the use of the tools of synthetic biology to create a set of individual yeast strains with systematic variation of substrate concentration. Finally, relatively standard techniques of metabolomics, flux measurements, and proteomics are creatively combined to obtain the basic Michaelis-Menten parameters for three enzymes in the carotenoid pathway.

This study would be a tour de force in its own right, but the fact that the techniques can be applied to other systems increases its importance. Though at present it is not clear how sophisticated the measurements can become (modern solution enzymology can dissect reaction kinetics and regulation much finer than these basic parameters), it is an exciting first step.

Specific comments:

- 1) The introduction does a nice job of putting this work in its historical context and giving credit to those who have made tentative forays into in vivo enzymology previously while pointing out the limitations that make those studies sui generis.
- 2) This leads me to the general comment that the paper is quite well-written and easy to follow.
- 3) I hate the term "fluxomics" and everyone else should too.
- 4) It is a pity that the authors have put some of the important material regarding the construction of the strains expressing the different substrate concentrations - the main innovation - into a supplemental figure (S1). Ideally, a modern paper, which will be viewed chiefly on-line, should contain no supplemental figures at all; in this case, S1 at least should become a figure in the main text.
- 5) Some measured parameters, such as the affinity of PaCrtB for GGPP, are so different from the in vitro values that they could indicate something interesting going on. The authors might comment more on this.
- 6) The authors provide a nice discussion on page 10 of the manuscript regarding the lack of saturation obtained for some of the studied enzymes. Their proposed explanations are sensible, but I am surprised they don't make more of the fact that their technique allows determination of whether or not a given enzyme is operating at saturation normally in vivo, under different cellular conditions. This could be as interesting to study as the kinetics - perhaps even more so.
- 7) I find no obvious technical concerns, but there is a question I think the authors might address in the Discussion: this is a lot of work to obtain just three basic kinetic parameters for three enzymes, even though they do leverage their strain construction to study other enzymes in the same pathway. Does the workload limit the applicability of the method, and how might things be improved?

Referee #2:

Castaño-Cerezo et al. develop and demonstrate an integrative framework for measuring enzyme kinetic properties in vivo. They expand the concept of in vivo apparent k_{cat} to affinity constants, and show how both can be directly measured using genetic modifications and standard metabolomics, and proteomics methods. I found the paper of interest and use. The paper contains useful figures, for example, Figure 1 is quite a masterpiece of effective science communication. I hope the comments below will help improve the paper further.

Major comments:

How were fluxes measured? The word "fluxomics" appears in the abstract (and figure 1), but there is no description of it in the methods or main text. There is one sentence about flux in page 14: "Phytoene and lycopene production fluxes were determined by multiplying their concentrations by the cell growth rate." If this is what "fluxomics" refers to, I suggest removing that term completely from the abstract, because it only refers to 2 reactions and thus not "omics" in essence. It is some sort of flux inference. Furthermore, it comes with a heavy assumption that is not clearly explained (namely that they are created irreversibly and accumulate in the cell without any further reaction/degradation). Also, there should be a referral to the methods section when flux is mentioned, especially in page 7: "phytoene concentration was too low for accurate quantification of the phytoene flux at low levels of PaCrtB". Without knowing which method was used this sentence is quite enigmatic. The method presented here relies heavily on the availability of quantitative in vivo measurements of flux and enzyme/metabolite concentrations. For it to be truly generalizable, one would need to be able to measure these for any reaction in the cell. However, in this proof-of-concept work, flux and metabolite concentrations were measured mainly using highly specific techniques (like the extracellular GGOH or the spectrophotometric methods for carotenoids) which are not generalizable. It would be helpful to discuss this caveat and how it can be overcome (e.g. using general metabolomics or fluxomics methods). It could greatly improve the clarity of the text, if the authors would more strictly separate between the strains where CrtE and CrtB are being manipulated. Originally, I was under the impression that both [E] and <S> will be measured in every single experiment, which could have been fine. However, in the actual fit the authors assumed that [E] is constant which meant CrtB was not changing in abundance. It took me a while to realize that the different expression levels of CrtB were only used to find an intermediate one that facilitates the other measurements within the bounds of their dynamic ranges. As this is intended to be a canonical example for any such in vivo kinetic assay, better clarity would be highly beneficial.

The authors allude to the usefulness of the obtained in vivo parameters, e.g. in the context of metabolic engineering. However, it is not very clear what can practically be done using the in vivo $K_{1/2}$. Since it was shown to differ substantially from in vitro values, what is the relevance of this finding and how can it be used to improve "industrial applications", as alluded to in the introduction.

Furthermore, V_{max} is much more robust than the $K_{1/2}$, since it doesn't suffer from the many problems that measuring internal metabolites entail. On the other hand, it might be useful to state somewhere that the value of $K_{1/2}$ can be found even without having absolute units for the flux, and without the need for proteomic measurements of the enzyme levels. The choice of an indirect method (extracellular GGOH) as a proxy for the substrate concentration (GGPP) for the main example is not ideal, so I am assuming it was difficult to find a better one. Nevertheless, the frequent switching between [GGOH] and [GGPP] in the text and figures is quite confusing and difficult to follow for a reader not swimming in the details. For example, at first glance figure 2b and 2c seem to show the same 6 strains, but apparently that is not the case when looking at the Y-axis scale ([GGOH]) which changes from 0-1 to 0-3. Furthermore, it would make more sense to plot the (estimated) [GGPP] concentration in 2C and 3B, as that is the more interesting value relevant for the enzyme in question.

Minor comments:

The linear fit in figure 2 seems to include an offset although it would make sense to constrain the offset to 0.

Page 7: "with the high ΔG^0 of this reaction" - probably the meaning was "very negative ΔG^0 "

The discussion could include ideas for what can be done in cases where some assumptions are not met, such as reversible reactions, reactions with co-factors that change concentration, or when the substrate is very differently distributed between compartments compared to the enzyme.

The link in page 16 ([https://github.com/MetaSys-LISBP/in_vivo_enzymatic_parameters.](https://github.com/MetaSys-LISBP/in_vivo_enzymatic_parameters)) doesn't work because the period is included in it.

Another point for discussion - one could claim that a better proof-of-concept for this method would have been a bacterium where compartments are a lesser problem. I'm wondering what the advantages of using yeast are.

Referee #3:

Castano-Cerezo et al., proposed an alternative approach to assess kinetic parameters of an enzyme in vivo. I agree that enzyme kinetics between test tube and in vivo physiological conditions can be significantly different. The authors collected quantitative data (proteome, flux, and metabolites) to investigate a synthetic carotenoid biosynthetic pathway, which I believe is an appropriate technique and dataset for calculating in vivo enzyme kinetics.

I do appreciate the authors efforts to collect and perform laborious measurements, however my concerns lie with the errors and integrity of the figures and texts exploring the approach. Here are some issues:

1. In Figure 2B, C. GGOH levels of the given engineered strains range from 0-3 $\mu\text{mol/gDCW}$ in panel 2C, but only from 0-1 $\mu\text{mol/gDCW}$ in panel 2B. To estimate GGPP concentration using GGOH levels, the regression should be valid for the full range of GGOH/GGPP. Were the six points in the two panels derived from the same strains? If yes, there is a discrepancy; if not, an explanation is needed as to why only a subset of strains with GGOH < 1 $\mu\text{mol/g DCW}$ was analyzed for regression.
2. On Page 5, the first sentence mentions Figure 2A, but it does not describe CrtB and CrtI.
3. On Page 7, line 17. references Supplementary Figure S2A, but there is only a single panel in Supplementary Figure S2.
4. The Western blots in Supplementary Figure S2 and S4 lack internal reference. Comparing protein expression across samples without a stable housekeeping protein (i.e., beta actin) is unreliable. Adding internal references is essential for accurate interpretation.
5. Could the authors provide a more detailed discussion regarding PaCrtI (Fig. 4B)? Apart from the low flux, could authors reason why the measured flux did not fit well with Michaelis-Menten kinetics? This could potentially highlight a limitation of the proposed assay and warrants further exploration and discussion.

We provide the revised manuscript with tracked changes along with this point-by-point response to the Referees' comments.

During the process of completing the data, we identified an error in the calculation of the GGPP concentration in the phytoene synthase (PaCrtB) experiments. This error affected the measured *in vivo* affinity of PaCrtB reported in the initial manuscript. However, the corrected results still indicate that the *in vivo* affinity remains different from the *in vitro* value. We apologize for this mistake, which has been corrected and does not affect the conclusions of the study. We have thoroughly checked all datasets and found no further errors in the manuscript.

Note: Referees' comments are in black, our responses are in blue.

Referee #1:

The paper describes what the authors call a generic approach for measuring enzymatic parameters (in this case, basic kinetic parameters) *in vivo*. It has been recognized for a long time that enzymes studies in dilute aqueous solution, the de facto condition for a hundred years, are in an environment unlike that inside a typical cell, where the molecular density approaches that of Times Square on New Year's Eve. (Ironically, the density inside a protein crystal - an environment scorned by classical enzymologists in the early days of structural biology as being completely artificial - is, in terms of crowding at least, more realistic.) As structural biology, driven by cryo-Electron Tomography, moves into the realm of *in vivo* observation, there is need for measurements of fundamental enzymatic parameters to do the same. That makes the work presented here potentially of great interest.

The authors use a number of steps to obtain their measurements, each of which has sufficient specific issues to render the claim of "a generic method" a bit exaggerated; nevertheless, the method is both clever and adaptable. The first step is the choice of organism: budding yeast is the best-studied eukaryote and one readily amenable to genetic manipulation and biochemical observation. The second, and cleverest, step is the use of the tools of synthetic biology to create a set of individual yeast strains with systematic variation of substrate concentration. Finally, relatively standard techniques of metabolomics, flux measurements, and proteomics are creatively combined to obtain the basic Michaelis-Menten parameters for three enzymes in the carotenoid pathway.

This study would be a tour de force in its own right, but the fact that the techniques can be applied to other systems increases its importance. Though at present it is not clear how sophisticated the measurements can become (modern solution enzymology can dissect reaction kinetics and regulation much finer than these basic parameters), it is an exciting first step.

Specific comments:

1) The introduction does a nice job of putting this work in its historical context and giving credit to those who have made tentative forays into *in vivo* enzymology previously while pointing out the limitations that make those studies *sui generis*.

We thank Referee #1 for their positive feedback on the introduction.

2) This leads me to the general comment that the paper is quite well-written and easy to follow.

We thank Referee #1 for their positive feedback on the clarity of the paper.

3) I hate the term "fluxomics" and everyone else should too.

Although we believe that the term "fluxomics" is appropriate for this study, as "targeted fluxomics" approaches are essential for quantifying fluxes even in a single reaction, we have removed the term from the revised manuscript.

4) It is a pity that the authors have put some of the important material regarding the construction of the stains expressing the different substrate concentrations - the main innovation - into a supplemental figure (S1). Ideally, a modern paper, which will be viewed chiefly on-line, should contain no supplemental figures at all; in this case, S1 at least should become a figure in the main text.

We agree with Referee #1 and have moved Figure S1 into the main text (now Figure 2).

5) Some measured parameters, such as the affinity of PaCrtB for GGPP, are so different from the in vitro values that they could indicate something interesting going on. The authors might comment more on this.

We appreciate the feedback. While our initial aim was to keep the paper concise, we acknowledge that it occasionally lacks detail and depth in its interpretation. In the revised manuscript, we have included additional hypotheses that could explain the deviation between in vivo and in vitro values:

"It is conceivable that, when these enzymes are bound to natural membranes, their apparent affinity for the substrate could differ from the in vitro environment. A second explanation concerns substrate availability. While classical in vitro enzyme assays involve homogeneous, highly diluted buffers, the intracellular environment is dense, heterogeneous and compartmentalized. In this first in vivo approach, variations in substrate concentrations between compartments were not considered explicitly, though it may have affected the values obtained for the parameters. For example, a higher concentration of phytoene inside lipid droplets compared with the rest of the membrane would reduce phytoene concentrations around CrtI enzymes and would thus limit substrate saturation. Enzyme localization could also affect measurements as conditions (pH, concentration of ions, etc.) vary between compartments. Iwata-Reuyl et al. found for instance that the measured activity of PaCrtB is 2,000 times lower in the absence of detergents (15), and Fournié et Truan observed that different heterologous CrtI expression systems produced different phytoene saturation patterns (36). In the case of the two P. ananas enzymes, the mechanisms that cause the absence of saturation may be different. For PaCrtI, the observed concentrations of phytoene (substrate) and lycopene (product) are either equivalent to or lower than those obtained with the two fungal CrtI enzymes. This suggests a true difference in the affinity for phytoene between the bacterial and fungal enzymes. For PaCrtB, given that the GGPP concentration in the cell is high (around 20 μ M), the lack of complete saturation is more likely due to a heterogeneous distribution of GGPP within the cell, with only a fraction of GGPP being in the vicinity of the enzyme. Dedicated studies will be required to clarify the effects of cell compartments on enzyme efficiency."

6) The authors provide a nice discussion on page 10 of the manuscript regarding the lack of saturation obtained for some of the studied enzymes. Their proposed explanations are sensible, but I am surprised they don't make more of the fact that their technique allows determination of whether or not a given enzyme is operating at saturation normally in vivo, under different cellular conditions. This could be as interesting to study as the kinetics - perhaps even more so.

We appreciate this comment. In response, we have added the following paragraph to the discussion section of the revised manuscript, where we also address the importance of determining saturation for applications in biotechnology:

“Our approach enables the determination of whether a given enzyme is operating at saturation in vivo under different cellular conditions. The tested enzymes displayed various saturation profiles, indicating that they may function under diverse conditions within the cell. Operating at full saturation (far above the K_M , the enzyme works at its maximum rate, being unaffected by changes in substrate concentration or minor environmental variations. This makes the enzymatic reaction highly robust in terms of product formation. Conversely, if an enzyme operates at substrate concentrations much lower than the K_M , any variation in substrate concentration will directly affect the production rate. This could maintain substrate homeostasis, potentially prevent toxic effects due to concentration changes. From a metabolic engineering perspective, achieving the optimal balance between high product formation and metabolic homeostasis is crucial. This balance can be attained by adjusting the levels of both the substrate-forming enzyme and the target enzyme. This process needs to be repeated for each enzyme, potentially under different saturation regimes. In biotechnology, our in vivo enzymology approach may thus guide metabolic engineering strategies to ensure that the overall pathway maintains the desired balance between production and cellular homeostasis, thereby ensuring greater stability and robustness of the engineered microbial strains at maximal production flux.”

7) I find no obvious technical concerns, but there is a question I think the authors might address in the Discussion: this is a lot of work to obtain just three basic kinetic parameters for three enzymes, even though they do leverage their strain construction to study other enzymes in the same pathway. Does the workload limit the applicability of the method, and how might things be improved?

The reviewer's observation is valid. Developing new techniques invariably takes time, and our in vivo enzymology approach is no exception. However, much like classical in vitro enzymology, we have gained sufficient expertise to establish a concise pipeline for efficiently designing and generating in vivo enzymatic data. We have included some guidance on this in the discussion section of the revised manuscript, where we also discuss how other kinetic parameters could be measured:

“We notably show how fine-tuning and balancing the expression of the substrate-producing enzyme and the enzyme under study yields datasets from which meaningful and reliable enzymatic parameters (K_M , V_{max} and k_{cat}) can be obtained. By including additional controlled steps, this method could be applied to a wider range of variables, such as inhibition and activation parameters (by modulating the pool of regulatory metabolites). This method could also benefit from dynamic data (time-course monitoring in response to metabolic or genetic perturbations). New formalisms will be required to account for in vivo conditions, notably the presence of products, similar enzyme and substrate concentrations, local concentration variations, and molecular fluxes within the cell.”

“As we advocate for the simplicity of our method, we would like to share some insights on its implementation in future studies. First, to maximize the output of genetic constructs, experimental design should be employed to compare various enzymes that catalyze the same reaction (whether mutants or from different species) or to target multiple enzymes in the same metabolic pathway, in this case phytoene synthase and phytoene desaturase. Second, the number of genetic constructs necessary to reach saturation (ideally 4 to 5) can be minimized by verifying the substrate production range early on. Third, in most cases, testing three enzyme concentrations has been sufficient to obtain a satisfactory saturation curve or at least to precisely estimate the K_M range. Lastly,

improvements can be made using high-throughput methods for the construction of plasmids and strains, for growth experiments and for samples preparation steps. The development of single cell metabolomics and proteomics approaches will also significantly increase the throughput of the strain characterization step in future studies."

Referee #2:

Castaño-Cerezo et al. develop and demonstrate an integrative framework for measuring enzyme kinetic properties in vivo. They expand the concept of in vivo apparent k_{cat} to affinity constants, and show how both can be directly measured using genetic modifications and standard metabolomics, and proteomics methods. I found the paper of interest and use. The paper contains useful figures, for example, Figure 1 is quite a masterpiece of effective science communication. I hope the comments below will help improve the paper further.

We thank Referee 2 for their positive feedback and interest in our work. We have revised the manuscript to address the comments below, which have indeed contributed to further improving it.

Major comments:

How were fluxes measured? The word "fluxomics" appears in the abstract (and figure 1), but there is no description of it in the methods or main text. There is one sentence about flux in page 14: "Phytoene and lycopene production fluxes were determined by multiplying their concentrations by the cell growth rate." If this is what "fluxomics" refers to, I suggest removing that term completely from the abstract, because it only refers to 2 reactions and thus not "omics" in essence. It is some sort of flux inference. Furthermore, it comes with a heavy assumption that is not clearly explained (namely that they are created irreversibly and accumulate in the cell without any further reaction/degradation). Also, there should be a referral to the methods section when flux is mentioned, especially in page 7: "phytoene concentration was too low for accurate quantification of the phytoene flux at low levels of PaCrtB". Without knowing which method was used this sentence is quite enigmatic.

We have replaced the term "fluxomics" with "flux measurement" throughout the manuscript. However, we would like to point out that, to date, no method exists for *direct* flux measurements. Therefore, fluxes are always estimated using "some sort of flux inference", i.e. involving various mathematical modeling approaches (isotopic calculations, mass balance, etc). In the methods, we have clarified how phytoene and lycopene production fluxes were estimated, outlining the underlying assumptions. We also reference previous work demonstrating that the method applied here yields results consistent with those obtained using targeted ^{13}C -fluxomics:

"Phytoene and lycopene are produced and accumulate in the cells, and their pools are continuously diluted by cell growth. Assuming an absence of degradation or reutilization of these end-products by the cell, phytoene and lycopene production fluxes are balanced solely by their dilution fluxes in the exponential growth phase, where cells are at metabolic steady-state. Thus, phytoene and lycopene production fluxes were determined by multiplying their concentrations by the cell growth rate. This flux calculation method provides results consistent with those obtained by targeted ^{13}C -fluxomics (54)."

Since "phytoene and lycopene production fluxes were determined by multiplying their concentrations by the cell growth rate", low precision in phytoene measurements (as observed at low metabolite concentration where the signal to noise ratio is low) directly translates into low

precision in the estimated fluxes. This is why we state that *“phytoene concentration was too low for accurate quantification of the phytoene flux at low levels of PaCrtB”*. Consequently, enzymatic parameters could not be estimated accurately (see the 95 % confidence intervals on K_M and V_{max} , Table EV3) from the set of strains with low CrtB activity (TEF1mut2 promoter), where low phytoene concentrations result in low precision for the phytoene production flux.

The method presented here relies heavily on the availability of quantitative *in vivo* measurements of flux and enzyme/metabolite concentrations. For it to be truly generalizable, one would need to be able to measure these for any reaction in the cell. However, in this proof-of-concept work, flux and metabolite concentrations were measured mainly using highly specific techniques (like the extracellular GGOH or the spectrophotometric methods for carotenoids) which are not generalizable. It would be helpful to discuss this caveat and how it can be overcome (e.g. using general metabolomics or fluxomics methods).

We partially agree with referee 2. The proposed strategy for *in vivo* enzymology (Figure 1) is generic. In this study, we employed established methodologies such as HPLC-MS and HPLC-UV for metabolite analysis, as well as Western Blot and proteomics for enzyme quantification. We believe these methods are widely accessible, either directly in most labs or through collaborative platforms, making the quantification of metabolites and enzymes broadly generalizable across various models. Nonetheless, we acknowledge that investigating a given enzyme requires a tailored setup, particularly for metabolite analysis, although techniques like HPLC-MS or GC-MS are likely to be effective in most scenarios. In the revised manuscript, we mention that *“the coverage of the metabolome and fluxome by current omics approaches is now high and is continuously increasing, which enables the application of the proposed approach to a broad range of enzymes”*.

It could greatly improve the clarity of the text, if the authors would more strictly separate between the strains where CrtE and CrtB are being manipulated. Originally, I was under the impression that both $[E]$ and $\langle S \rangle$ will be measured in every single experiment, which could have been fine. However, in the actual fit the authors assumed that $[E]$ is constant which meant CrtB was not changing in abundance. It took me a while to realize that the different expression levels of CrtB were only used to find an intermediate one that facilitates the other measurements within the bounds of their dynamic ranges. As this is intended to be a canonical example for any such *in vivo* kinetic assay, better clarity would be highly beneficial.

We agree with referee 2 and have clarified this point in the revised manuscript. While $[S]$ was measured in each experiment, the concentration of CrtB was not. However, we verified by western blot and MS-based proteomics that the expression level of $[E]$ was stable across the different strains. For clarity, we have conducted simulations to illustrate the importance of tuning the level of the studied enzyme (Fig 4B). We have also performed new experiments demonstrating how varying CrtB abundance affects the saturation profiles (Fig 4C). Finally, using GGPP alone, instead of a mix of GGPP and GGOH as substrate proxy, has further clarified our result section.

The authors allude to the usefulness of the obtained *in vivo* parameters, e.g. in the context of metabolic engineering. However, it is not very clear what can practically be done using the *in vivo* $K_{1/2}$. Since it was shown to differ substantially from *in vitro* values, what is the relevance of this finding and how can it be used to improve "industrial applications", as alluded to in the introduction.

We agree with Referee #2. In response, we provide details on how our findings could inform metabolic engineering strategies for biotechnological applications. Specifically, we expand upon this concept to encompass other enzymes, distinguishing between the importance of having enzymatic

reactions operating at V_{max} versus those operating within the $K_{1/2}$ range. We have added the following paragraph in the discussion:

“Our approach enables the determination of whether a given enzyme is operating at saturation in vivo under different cellular conditions. The tested enzymes displayed various saturation profiles, indicating that they may function under diverse conditions within the cell. Operating at full saturation (far above the $K_{1/2}$, the enzyme works at its maximum rate, being unaffected by changes in substrate concentration or minor environmental variations. This makes the enzymatic reaction highly robust in terms of product formation. Conversely, if an enzyme operates at substrate concentrations much lower than the $K_{1/2}$, any variation in substrate concentration will directly affect the production rate. This could maintain substrate homeostasis, potentially prevent toxic effects due to concentration changes. From a metabolic engineering perspective, achieving the optimal balance between high product formation and metabolic homeostasis is crucial. This balance can be attained by adjusting the levels of both the substrate-forming enzyme and the target enzyme. This process needs to be repeated for each enzyme, potentially under different saturation regimes. In biotechnology, our in vivo enzymology approach may thus guide metabolic engineering strategies to ensure that the overall pathway maintains the desired balance between production and cellular homeostasis, thereby ensuring greater stability and robustness of the engineered microbial strains at maximal production flux.”

Furthermore, V_{max} is much more robust than the $K_{1/2}$, since it doesn't suffer from the many problems that measuring internal metabolites entail. On the other hand, it might be useful to state somewhere that the value of $K_{1/2}$ can be found even without having absolute units for the flux, and without the need for proteomic measurements of the enzyme levels.

We thank Referee #2 for this comment. We have incorporated this insight into the discussion section of the manuscript, as follows: *“Similarly, estimating the absolute values can be achieved using only relative flux values, without requiring absolute concentration of enzymes.”*

The choice of an indirect method (extracellular GGOH) as a proxy for the substrate concentration (GGPP) for the main example is not ideal, so I am assuming it was difficult to find a better one. Nevertheless, the frequent switching between [GGOH] and [GGPP] in the text and figures is quite confusing and difficult to follow for a reader not swimming in the details. For example, at first glance figure 2b and 2c seem to show the same 6 strains, but apparently that is not the case when looking at the Y-axis scale ([GGOH]) which changes from 0-1 to 0-3. Furthermore, it would make more sense to plot the (estimated) [GGPP] concentration in 2C and 3B, as that is the more interesting value relevant for the enzyme in question.

We agree that the use of both GGOH and GGPP data can be confusing to the readers. We have replaced the GGOH data with direct measurements of GGPP in all strains varying CrtB levels (see section *In vivo enzymatic parameters of phytoene synthase from Pantoea ananas*). These new results confirm our initial conclusions, improve clarity, and overall strengthen the manuscript.

Minor comments:

The linear fit in figure 2 seems to include an offset although it would make sense to constrain the offset to 0.

We have removed this panel as GGOH is no longer used as a proxy for GGPP.

Page 7: "with the high ΔG^0 of this reaction" - probably the meaning was "very negative ΔG^0 "

We have changed the sentence as suggested by the referee.

The discussion could include ideas for what can be done in cases where some assumptions are not met, such as reversible reactions, reactions with co-factors that change concentration, or when the substrate is very differently distributed between compartments compared to the enzyme.

We have included the following points in the discussion.

“New formalisms will be required to account for in vivo conditions, notably the presence of products, similar enzyme and substrate concentrations, local concentration variations, and molecular fluxes within the cell.”

“For PaCrtB, given that the GGPP concentration in the cell is high (around 20 μM), the lack of saturation is more likely due to a heterogeneous distribution of GGPP within the cell, with only a fraction of GGPP being in the vicinity of the enzyme. Dedicated studies will be required to clarify the effects of cell compartments on enzyme efficiency.”

The link in page 16 ([https://github.com/MetaSys-LISBP/in_vivo_enzymatic_parameters.](https://github.com/MetaSys-LISBP/in_vivo_enzymatic_parameters)) doesn't work because the period is included in it.

The link is corrected in the revised version.

Another point for discussion - one could claim that a better proof-of-concept for this method would have been a bacterium where compartments are a lesser problem. I'm wondering what the advantages of using yeast are.

We agree with the reviewer's observation. This question indeed arose at the start of our experimental design. We collectively felt that if a method proves to be effective for assessing in vivo enzymatic parameters, it should be applicable across different model organisms. While a bacterial model might be simpler, especially for soluble substrates that do not partition within the cell, the carotene pathway is more complex, involving enzymes and substrates that are predominantly membrane-bound. Investigating these enzymes in vitro is particularly challenging, so demonstrating the value of in vivo enzymology using such a complex case underscores the intrinsic value of the proposed approach.

Given our extensive experience with the metabolic engineering of carotenoids in yeast—along with access to strains, plasmids, and expertise—we chose yeast as our model organism. In the future, we plan to apply the proposed methodology to other microorganisms, including the bacterium *Escherichia coli*.

Referee #3:

Castano-Cerezo et al., proposed an alternative approach to assess kinetic parameters of an enzyme in vivo. I agree that enzyme kinetics between test tube and in vivo physiological conditions can be

significantly different. The authors collected quantitative data (proteome, flux, and metabolites) to investigate a synthetic carotenoid biosynthetic pathway, which I believe is an appropriate technique and dataset for calculating in vivo enzyme kinetics.

I do appreciate the authors efforts to collect and perform laborious measurements, however my concerns lie with the errors and integrity of the figures and texts exploring the approach. Here are some issues:

1. In Figure 2B, C. GGOH levels of the given engineered strains range from 0-3 $\mu\text{mol/gDCW}$ in panel 2C, but only from 0-1 $\mu\text{mol/gDCW}$ in panel 2B. To estimate GGPP concentration using GGOH levels, the regression should be valid for the full range of GGOH/GGPP. Were the six points in the two panels derived from the same strains? If yes, there is a discrepancy; if not, an explanation is needed as to why only a subset of strains with GGOH < 1 $\mu\text{mol/g DCW}$ was analyzed for regression.

We have now measured GGPP for all strains and no longer use GGOH as a proxy for GGPP concentration. Specifically, we have produced the three saturation curves (at low, medium, high levels of CrtB) by directly measuring the GGPP concentration (Figure 3C). We believe this addition strengthens our results and clarifies the manuscript.

2. On Page 5, the first sentence mentions Figure 2A, but it does not describe CrtB and CrtI.

We have corrected this error.

3. On Page 7, line 17. references Supplementary Figure S2A, but there is only a single panel in Supplementary Figure S2.

We have corrected this typo.

4. The Western blots in Supplementary Figure S2 and S4 lack internal reference. Comparing protein expression across samples without a stable housekeeping protein (i.e., beta actin) is unreliable. Adding internal references is essential for accurate interpretation.

As suggested by Referee 3, we have performed new Western Blots with an internal standard (beta actin) to compare protein levels. Results confirm our initial conclusions.

5. Could the authors provide a more detailed discussion regarding PaCrtI (Fig. 4B)? Apart from the low flux, could authors reason why the measured flux did not fit well with Michaelis-Menten kinetics? This could potentially highlight a limitation of the proposed assay and warrants further exploration and discussion.

We agree with Referee #3. In response, we provide additional hypotheses on the reason why saturation is not achieved. This point was also highlighted by reviewer 1, and we have revised the discussion accordingly.

"We demonstrate that substrate concentrations can be varied in vivo over a wide range. Here, the GGPP concentration was varied by a factor 167 and the phytoene concentration by a factor 430, allowing clear enzyme saturation curves to be observed for the two fungal phytoene desaturases (XdCrtI and BtCrtI). However, while the range of substrate concentrations explored was significantly

wider than those used in *in vitro* studies of the same enzymes (Table EV4) (16, 21), complete saturation was not achieved for PaCrtB or PaCrtI. It is conceivable that, when these enzymes are bound to natural membranes, their apparent affinity for the substrate could differ from the *in vitro* environment. A second explanation concerns substrate availability. While classical *in vitro* enzyme assays involve homogeneous, highly diluted buffers, the intracellular environment is dense, heterogeneous and compartmentalized. In this first *in vivo* approach, variations in substrate concentrations between compartments were not considered explicitly, though it may have affected the values obtained for the parameters. For example, a higher concentration of phytoene inside lipid droplets compared with the rest of the membrane would reduce phytoene concentrations around CrtI enzymes and would thus limit substrate saturation. Enzyme localization could also affect measurements as conditions (pH, concentration of ions, etc.) vary between compartments. Iwata-Reuyl *et al.* found for instance that the measured activity of PaCrtB is 2,000 times lower in the absence of detergents (15), and Fournié *et Truan* observed that different heterologous CrtI expression systems produced different phytoene saturation patterns (36). In the case of the two *P. ananas* enzymes, the mechanisms that cause the absence of saturation may be different. For PaCrtI, the observed concentrations of phytoene (substrate) and lycopene (product) are either equivalent to or lower than those obtained with the two fungal CrtI enzymes. This suggests a true difference in the affinity for phytoene between the bacterial and fungal enzymes. For PaCrtB, given that the GGPP concentration in the cell is high (around 20 μ M), the lack of complete saturation is more likely due to a heterogeneous distribution of GGPP within the cell, with only a fraction of GGPP being in the vicinity of the enzyme. Dedicated studies will be required to clarify the effects of cell compartments on enzyme efficiency.”

Dr. Gilles Truan
Toulouse Biotechnology Institute
135 Avenue de Rangueil
Toulouse 31077
France

3rd Sep 2024

Re: EMBOJ-2024-117139R
Combining systems and synthetic biology for in vivo enzymology

Dear Dr. Truan,

Thank you for submitting your revised manuscript for our consideration. As you will see from the comments copied below, all three original referees were fully satisfied with the revisions. We shall therefore be happy to accept the study for The EMBO Journal, following incorporation of several remaining editorial issues:

- Please adjust the format of the reference list and of the in-text citations according to EMBO Journal format (alphabetical order, author name et al + year...). When doing so, make sure to also adjust the format for citation of preprints as specified in our author guidelines:

The citation in the text should be: "(preprint: NAME1 et al, YEAR)"

The citation in the reference list: "Author NAME1, Author NAME2, ... (YEAR) article title. bioRxiv/ResearchSquare doi: XXX"

- On the abstract page of the manuscript, please include 4-5 general keyword terms to enhance searchability.

- Please note that Materials and Methods need to be described in the main text using our 'Structured Methods' format (for detail, see <https://www.embopress.org/page/journal/14693178/authorguide#structuredmethods>). The in-text "Methods" section should contain method and protocol descriptions (ideally using a step-by-step protocol format to facilitate adoption of the methodologies across labs), while all key reagents, experimental models, software and relevant equipment - including their sources and relevant identifiers - should be listed in a separately uploaded Reagents and Tools Table, a template for which can be downloaded from the above section of our Author Guidelines, but is also attached to this message.

- For all Expanded View Tables, make sure to remove their legends from the main text, and instead include each legend in the respective XLSX file, in a separate "Legend" tab. For Dataset EV1 supposed to contain plasmid sequences and annotations, please consider using a more easily accessible file type, if at all possible.

- As we are switching from a free-text author contribution statement towards a more formal statement based on Contributor Role Taxonomy (CRediT) terms, please remove the present Author Contribution section and instead specify each author's contribution(s) directly in the Author Information page of our submission system during upload of the final manuscript. See <https://casrai.org/credit/> for more information.

- Please rename the Competing Interests section into "Disclosure and Competing Interests Statement", in accordance with our updated Guide to Authors (<https://www.embopress.org/competing-interests>)

- Please move the funding information, currently included in a separate section, into the Acknowledgement section.

- Please reorganize the uploaded Source Data so that one individual ZIP archive is provided per main figure, while Source Data for EV Figures is combined in one single ZIP.

- Finally, please provide suggestions for a short 'blurb' text prefacing and summing up the conceptual aspect of the study in two sentences (max. 250 characters), followed by 3-5 one-sentence 'bullet points' with brief factual statements of key results of the paper; they will form the basis of an editor-written 'Synopsis' accompanying the online version of the article. Please also upload a synopsis image, which can be used as a "visual title" for the synopsis section of your paper. The image (maybe based on Figure 1?) should be in PNG or JPG format, and please make sure that it remains in the modest dimensions of (exactly) 550 pixels wide and 300-600 pixels high.

I am therefore inviting you to a final round of formal revision, solely to allow you to make these modifications and upload all revised files. Once we will have received them, we should hopefully be ready to swiftly proceed with publication of the study!

Yours sincerely,

Hartmut Vodermaier

*** PLEASE NOTE: All revised manuscripts are subject to initial checks for completeness and adherence to our formatting guidelines. Revisions may be returned to the authors and delayed in their editorial re-evaluation if they fail to comply to the following requirements (see also our Guide to Authors for further information):

9) To facilitate reproducibility and cross-laboratory adoption of methodologies, please structure the Materials & Methods section as outlined in our guide to authors, including a completed Reagents and Tools Table that can be downloaded from our author guidelines as well (<https://www.embopress.org/page/journal/14602075/authorguide#structuredmethods>).

10) Digital image enhancement is acceptable practice, as long as it accurately represents the original data and conforms to community standards. If a figure has been subjected to significant electronic manipulation, this must be clearly noted in the figure legend and/or the 'Materials and Methods' section. The editors reserve the right to request original versions of figures and the original images that were used to assemble the figure. Finally, we generally encourage uploading of numerical as well as gel/blot image source data; for details see: embopress.org/page/journal/14602075/authorguide#sourcedata

At EMBO Press, we ask authors to provide source data for the main manuscript figures. Our source data coordinator will contact you to discuss which figure panels we would need source data for and will also provide you with helpful tips on how to upload and organize the files.

Further information is available in our Guide For Authors:

In the interest of ensuring the conceptual advance provided by the work, we recommend submitting a revision within 3 months (2nd Dec 2024). Please discuss the revision progress ahead of this time with the editor if you require more time to complete the revisions. Use the link below to submit your revision:

Link Not Available

Referee #1:

The authors have satisfied my concerns. The paper is much improved and, more importantly, more useful.

Referee #2:

My concerns were fully addressed. I congratulate the authors for their contribution to the community

Referee #3:

The authors have successfully addressed all previous concerns and conducted additional experiments to resolve any remaining errors and ambiguities. These efforts have improved the manuscript, and I now recommend the revised version for publication

Dr. Gilles Truan
Toulouse Biotechnology Institute
135 Avenue de Rangueil
Toulouse 31077
France

12th Sep 2024

Re: EMBOJ-2024-117139R1
Combining systems and synthetic biology for in vivo enzymology

Dear Dr. Truan,

Thank you for submitting your final revised manuscript for our consideration. I am pleased to inform you that we have now accepted it for publication in The EMBO Journal.

Yours sincerely,

Hartmut Vodermaier
